# Imaging the response to DNA damage in heterochromatin domains reveals core principles of heterochromatin maintenance

Anna Fortuny[1], Audrey Chansard[1], Pierre Caron[1], Odile Chevallier[1], Olivier Leroy [2], Olivier Renaud [2] & Sophie E. Polo [1✉]

Heterochromatin is a critical chromatin compartment, whose integrity governs genome stability and cell fate transitions. How heterochromatin features, including higher-order chromatin folding and histone modifications associated with transcriptional silencing, are maintained following a genotoxic stress challenge is unknown. Here, we establish a system for targeting UV damage to pericentric heterochromatin in mammalian cells and for tracking the heterochromatin response to UV in real time. We uncover profound heterochromatin compaction changes during repair, orchestrated by the UV damage sensor DDB2, which stimulates linker histone displacement from chromatin. Despite massive heterochromatin unfolding, heterochromatin-specific histone modifications and transcriptional silencing are maintained. We unveil a central role for the methyltransferase SETDB1 in the maintenance of heterochromatic histone marks after UV. SETDB1 coordinates histone methylation with new histone deposition in damaged heterochromatin, thus protecting cells from genome instability. Our data shed light on fundamental molecular mechanisms safeguarding higher-order chromatin integrity following DNA damage.

[1] Epigenetics and Cell Fate Centre, UMR7216 CNRS, Université de Paris, Paris, France. [2] Cell and Tissue Imaging Facility, UMR3215 PICT-IBiSA, Institut Curie, Paris, France. ✉email: sophie.polo@univ-paris-diderot.fr

Eukaryotic cell identity and function are governed by the epigenetic information stored in the form of chromatin inside the cell nucleus, where DNA wraps around histone proteins[1]. This information encompasses multiple layers of regulation, from histone modifications[2] and histone variants[3], up to higher-order folding of the chromatin fiber into nuclear domains[4], which, in concert, control gene expression. Among higher-order chromatin domains, heterochromatin covers a significant fraction of metazoan genomes[5,6] and plays a central role in the maintenance of genome stability. Highly concentrated at pericentromeric and subtelomeric regions, heterochromatin is indeed crucial for chromosome segregation and integrity, and alterations of heterochromatin features are commonly associated with aging and cancer[7]. Furthermore, heterochromatin formation is instrumental for silencing repetitive elements and preventing their illegitimate recombination[7,8]. Heterochromatin silencing is mediated by specific patterns of histone post-translational modifications[7,9]. For instance, pericentric heterochromatin domains[10] carry a distinct chromatin signature, including trimethylation on H3 lysine 9 (H3K9me3) and on H4 lysine 20 (H4K20me3)[11], which contribute to epigenetic silencing of major satellite repeats. H3K9me3 heterochromatin also plays a pivotal role in defining cell identity by silencing lineage-specific genes during development[12,13].

Considering the profound influence of heterochromatin on genome stability and cell fate transitions, much effort has been devoted to understanding how heterochromatin domains are established during development and maintained through cell divisions[9,14]. One of the most persistent challenges to heterochromatin maintenance is the response to DNA damage, which can arise at any time, anywhere in the genome[15,16] and poses a major threat to epigenome stability[17,18]. Indeed, substantial rearrangements affect chromatin during the repair response, including histone exchange and chromatin mobility[17,19], changes in histone post-translational modifications[20] and alterations in chromatin compaction[21–27]. These rearrangements are accompanied by transient changes in chromatin transcriptional activity[28–30]. The destabilization of chromatin organization upon genotoxic stress is followed by a restoration of chromatin structure[31,32]. However, our knowledge of this fundamental process is still largely incomplete and little is known about the maintenance of higher-order heterochromatin domains following DNA damage.

Moreover, due to high compaction and to the abundance of repeated sequences prone to ectopic recombination, heterochromatin represents a challenging environment for the DNA damage response. Heterochromatic regions indeed pose a barrier to DNA damage signalling[33] and repair, as described for nucleotide excision repair (NER)[34–36], DNA double-strand break (DSB) repair[37–39] and mismatch repair[40] in mammalian cells. In line with this, higher mutation rates are found in heterochromatin in human cancer genomes[36,41].

In recent years, exciting progress has been made in understanding how DNA damage repair proceeds in heterochromatin, as mostly studied in response to DNA breaks[42]. In drosophila and mouse cells, DSBs elicit a decompaction of pericentric heterochromatin and relocate to the periphery of heterochromatin domains for the completion of recombinational repair, which is thought to prevent illegitimate recombination between pericentromeric repeats[43–46]. However, beyond the restoration of genome integrity, the mechanisms underlying the maintenance of heterochromatic features during the repair response remain uncharacterized (reviewed in[47]). In particular, how heterochromatin compaction and silencing histone marks are preserved following DNA damage is still unknown, and whether they are maintained in a concerted manner also remains elusive.

Here, we explore these mechanisms by inflicting UV damage to pericentric heterochromatin domains in mammalian cells. We reveal that heterochromatin-specific histone marks and transcriptional silencing are maintained in damaged heterochromatin, despite massive heterochromatin unfolding. We demonstrate that heterochromatin unfolding is driven by the UV damage sensor DNA damage-binding protein 2 (DDB2), which stimulates linker histone displacement from damaged chromatin. Our findings also unveil a tight cooperation between histone chaperones and histone-modifying enzymes in the maintenance of heterochromatic histone marks following UV damage.

## Results

**A mammalian cellular model to track the heterochromatin response to UV damage.** In order to study heterochromatin maintenance in response to DNA damage, we first established an appropriate cellular model where heterochromatin domains could be easily distinguished and where DNA repair events and histone deposition into chromatin could be tracked. For this purpose, we selected NIH/3T3 mouse embryonic fibroblasts, characterized by a clustering of pericentric heterochromatin domains into chromocenters[48] (Fig. 1a), and we focused on the cell response to UVC damage (Supplementary Fig.1a). Noteworthy, mouse fibroblasts express the UV damage sensor DDB2 at very low levels, which impairs both UVC damage repair[49] and repair-coupled histone dynamics[50,51]. To overcome these defects, NIH/3T3 stable cell lines were engineered to ectopically express GFP-tagged human DDB2 (GFP-hDDB2), which did form a complex with mouse DNA damage-binding protein 1, as expected (Supplementary Fig. 1a–c). These cells also stably express SNAP-tagged H3.3, which allows specific tracking of newly synthesized H3.3 histones[52] (Supplementary Fig. 1a, see Supplementary Fig. 1d–f for a complete characterization of the cell lines). The ectopic expression of DDB2 and H3.3 did not affect pericentric heterochromatin organization as judged by immunostaining for H3K9me3 and heterochromatin protein 1 α (HP1α) (Fig. 1a). We verified that GFP-hDDB2 expression rescued UVC damage repair and associated histone dynamics in mouse cells, by analyzing the recruitment of the NER factor xeroderma pigmentosum complementation group B (XPB) and the deposition of newly synthesized H3.3 histones at sites of UVC damage (Supplementary Fig. 1g).

**Heterochromatin integrity is maintained in response to UV damage.** Using the mammalian cellular model described above, we first assessed the importance of heterochromatin integrity for the cellular response to UV damage. We impaired heterochromatin integrity by knocking down the histone methyltransferases SUV39H1 and 2 (Suppressor of Variegation 3–9 Homolog1/2), which are the main drivers of H3K9me3 in pericentric heterochromatin[53] (Fig. 1a), and tested the ability of SUV39H1/2-depleted cells to survive UVC damage. We observed that SUV39H1/2 knockdown led to a modest, albeit significant, decrease in cell survival to global UVC irradiation (Fig. 1a). Loss of heterochromatin integrity thus correlates with reduced cell viability following UV damage.

To determine whether heterochromatin integrity was preserved following a genotoxic stress challenge, we developed an innovative approach for targeting UVC damage to pericentric heterochromatin domains in live cells and for tracking the response to heterochromatin damage in real time. We employed the live-cell DNA stain Hoechst 33258 to visualize chromocenters in mouse cells and then inflicted UVC damage specifically to chromocenters of interest by using a UVC laser coupled to a confocal microscope (Fig. 1b, see Supplementary Fig. 2 for a

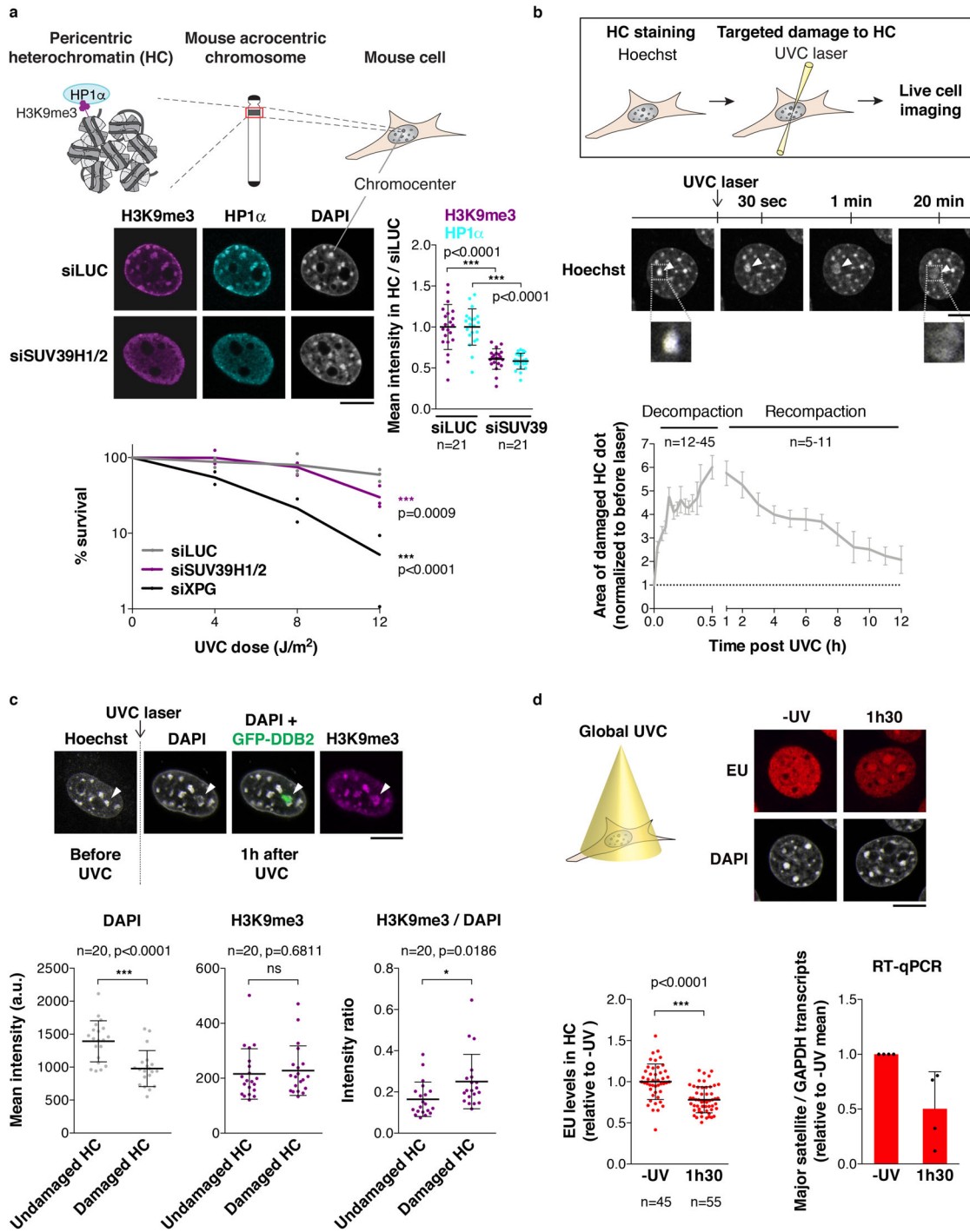

**Fig. 1 Heterochromatin integrity is maintained in response to UV damage. a** Schematic representation of pericentric heterochromatin domains in mouse cells and delocalization of heterochromatin marks (H3K9me3, HP1α) upon knockdown of SUV39H1/2 methyltransferases in NIH/3T3 GFP-DDB2 cells. Clonogenic survival of the same cell line treated with the indicated siRNAs (siLUC, negative control; siXPG, positive control) and exposed to global UVC irradiation. **b** Technical approach for targeting UVC damage to pericentric heterochromatin domains (HC) in live murine cells. Heterochromatin compaction changes upon UVC laser micro-irradiation are analyzed by live imaging in NIH/3T3 GFP-DDB2 cells stained with Hoechst. White arrowheads point to UVC-damaged heterochromatin domains. **c** H3K9me3 levels in damaged heterochromatin (white arrowheads) analyzed by immunofluorescence in NIH/3T3 GFP-DDB2 cells 1 h after UVC laser micro-irradiation. Scatter plots represent DAPI and H3K9me3 levels measured on reconstructed 3D images in damaged heterochromatin (HC) domains compared to undamaged heterochromatin in the same nucleus. **d** Heterochromatin transcription analyzed 1h30 after global UVC damage in NIH/3T3 GFP-DDB2 cells by EU staining (fluorescence images and left graph) and by RT-qPCR for major satellite transcripts (right graph). Data are presented as mean values ± SD (SEM for (**b**) panel only) from four experiments ((**d**), right graph) or from n cells scored in at least three independent experiments. Statistical significance is calculated via two-sided Student's *t* test with Welch's correction when necessary (**a**, **c**, **d**). Comparisons of clonogenic survival in (**a**) are based on non-linear regression with a polynomial quadratic model. a.u. arbitrary units. All microscopy images are confocal sections. Scale bars, 10 μm. Zoomed in views of heterochromatin domains (×2.6). Source data are provided as a Source Data file.

characterization of UVC laser damage combined with Hoechst staining). Using this approach, we observed a pronounced and rapid decompaction of damaged heterochromatin within minutes after UVC laser damage, reaching a maximum (up to sixfold) 30 min to 1 h after irradiation (Fig. 1b and Supplementary Movie 1). Heterochromatin decompaction was restricted to UVC-damaged chromocenters (Supplementary Fig. 3a), and was further confirmed by DNA-fluorescence in situ hybridization (DNA-FISH) analysis of mouse major satellite sequences (Supplementary Fig. 3b). UV-induced heterochromatin decompaction was not restricted to a specific cell cycle stage as it was observed both in and outside S-phase (Supplementary Fig. 3c). Importantly, damaged heterochromatin decompaction was followed by a slower recompaction phase taking several hours, which restored heterochromatin compaction close to its original state (Fig. 1b and Supplementary Movie 2). Furthermore, immunostaining for H3K9me3 in cells fixed 1 h after UVC laser damage revealed that damaged heterochromatin decompaction was not associated with a reduction of this heterochromatin-specific histone mark, which instead appeared slightly increased on damaged chromocenters (Fig. 1c). Similar results were obtained when staining for H4K20me3 (Supplementary Fig. 3d). The observed increase of silencing marks was restricted to damaged chromocenters, with no detectable increase of H3K9me3 in damaged euchromatin regions (Supplementary Fig. 3e). In addition, a modest but reproducible increase in H3K9me3 levels was detected by western blot on total extracts from cells exposed to global UVC irradiation (Supplementary Fig. 3f, g), which confirms the above findings and excludes the possibility of increased H3K9me3 detection due to increased antibody accessibility in decompacted heterochromatin. Kinetic analyses showed a gradual increase in total H3K9me3 levels up to 3 h post UV without any detectable drop at early time points (Supplementary Fig. 3g). Together, these experiments demonstrate that heterochromatin-specific histone marks are maintained, and even slightly increased, in UVC-damaged heterochromatin, rather than removed and subsequently re-established. In line with these findings, damaged heterochromatin decompaction was not accompanied by a burst of aberrant transcription. Indeed, the staining of nascent transcripts with Ethynyl–Uridine (EU) and the quantification of their levels in heterochromatin domains before and after UV irradiation revealed that transcription was even further reduced in UV-damaged heterochromatin (Fig. 1d, left graph). This UV-induced transcriptional arrest in heterochromatin was confirmed by RT-qPCR of pericentric major satellite transcripts (Fig. 1d, right graph). From these observations, we conclude that UVC damage challenges heterochromatin integrity and that maintenance mechanisms operate to restore heterochromatin compaction and to reinforce heterochromatin-specific histone marks and heterochromatin silencing following UV damage.

**The UV damage sensor DDB2 regulates heterochromatin compaction**. To characterize the mechanisms underlying heterochromatin maintenance following UVC damage, we first sought to identify the molecular trigger for damaged heterochromatin decompaction. For this, we examined the potential contribution of Poly(ADP-ribosyl)ation (PARylation), which was involved in damaged chromatin decompaction in several studies[24,27,54]. However, the chemical inhibition of Poly(ADP-ribose) Polymerase (PARP) had no measurable impact on heterochromatin decompaction following UV damage (Supplementary Fig. 4a). We next focused on the UV damage sensor DDB2, whose binding to chromatin was shown to promote histone redistribution and chromatin relaxation in human cells[23,51]. Noteworthy, we observed decompaction of UV-damaged

heterochromatin domains only in the engineered cell line expressing hDDB2 and not in the parental mouse cell line (DDB2 deficient) (Fig. 2a), supporting the idea that DDB2 is required for heterochromatin decompaction following UVC damage.

To directly test whether DDB2 could drive heterochromatin decompaction, we tethered GFP-hDDB2 to mouse pericentric heterochromatin in the absence of DNA damage by co-expressing catalytically dead Cas9 (dCas9) fused to a GFP nanobody and a guide RNA targeting major satellite repeats[55] (Fig. 2b and Supplementary Fig. 4b, c). DDB2 tethering led to substantial changes in the shape and size of pericentric heterochromatin domains, which were enlarged and less spherical compared to control cells, indicative of a decompaction of pericentric heterochromatin domains. This effect was specific to DDB2 tethering as it was not observed upon targeting of another early NER factor, Xeroderma Pigmentosum complementation group C (XPC), to chromocenters (Fig. 2c and Supplementary Fig. 4d). Consistent with this, heterochromatin decompaction following UV damage still occurred upon knockdown of XPC (Supplementary Fig. 4e), arguing that heterochromatin decompaction does not rely on UV damage processing. Further supporting this conclusion, the decompaction of heterochromatin observed upon dCas9-mediated tethering of DDB2 occurs without the recruitment of UV damage processing factor XPB (Supplementary Fig. 4f).

Noteworthy, when we induced the release of tethered DDB2 from major satellite repeats with the anti-Cas9 bacteriophage protein AcrIIA4, thus mimicking the release of DDB2 from damaged chromatin that occurs during repair progression[56], the typical size and shape of chromocenters were restored, showing that DDB2 release allows pericentric heterochromatin recompaction (Fig. 2d and Supplementary Fig. 4g). Collectively, these findings establish that the UV damage sensor DDB2 is both necessary and sufficient for driving changes in heterochromatin compaction following UVC damage.

**The UV damage sensor DDB2 stimulates linker histone displacement from damaged chromatin**. We next sought to characterize the mechanisms underlying DDB2-mediated changes in heterochromatin compaction following UV damage. Given that DDB2 does not harbour known chromatin remodelling activity or motifs, it is expected to control heterochromatin compaction indirectly. Supporting this idea, DDB2 recruitment to and release from damaged heterochromatin occurred at least 30 min earlier than the changes in heterochromatin compaction (Supplementary Fig. 4h). The indirect effect of DDB2 on chromatin compaction could be mediated by the recruitment of chromatin remodellers but our loss-of-function approaches against candidate remodelling factors did not recapitulate the effect of DDB2 depletion on chromatin unfolding[51]. We thus explored the alternative hypothesis that DDB2 could alter chromatin folding via the release of factors involved in chromatin compaction such as linker histones[57], which are key for constitutive heterochromatin maintenance[58]. For this, we focused on two somatic linker histone H1 variants, H1.0 and H1.4, because they display strong chromatin compaction properties in vitro and localize to pericentric heterochromatin in mouse fibroblasts[59,60] (Fig. 3a and Supplementary Fig. 4i). In NIH/3T3 GFP-DDB2 cells expressing mCherry-tagged linker histone variants, we observed a local depletion of linker histones H1.0 and H1.4 at sites of UVC laser damage (Fig. 3a and Supplementary Fig. 4i). The reduction in H1 levels was not merely reflecting chromatin decompaction at damage sites since the levels of core histones H3.3 and H2B were not reduced to the same extent (Fig. 3a and Supplementary Fig. 4i), even at early time points post irradiation (10 min), when the contribution of core histone deposition is negligible[50]. We

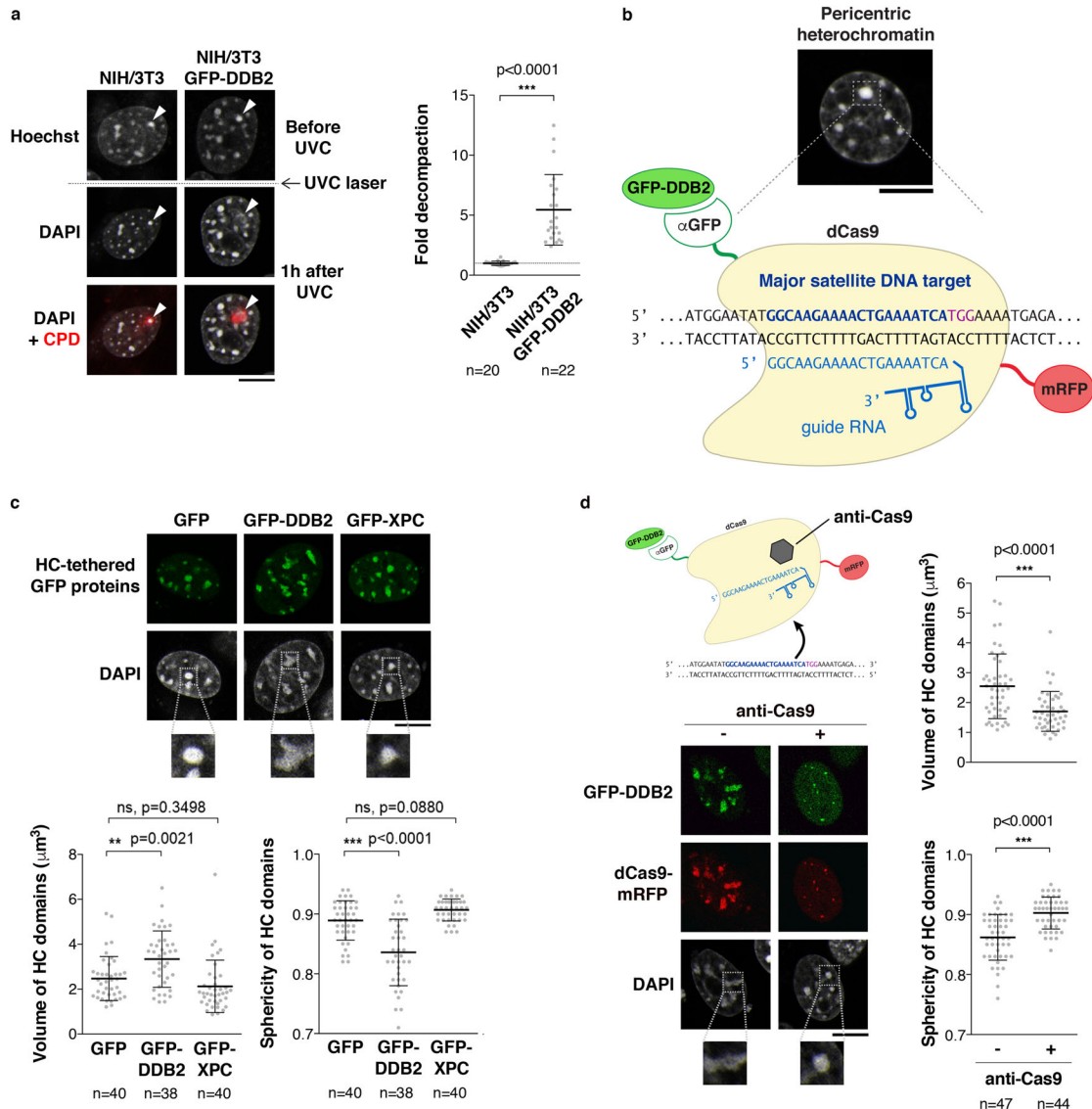

**Fig. 2 The UV damage sensor DDB2 regulates heterochromatin compaction. a** Decompaction of damaged pericentric heterochromatin domains (white arrowheads) 1 h after UVC laser micro-irradiation analyzed by live imaging in the indicated cell lines. CPD staining in fixed cells highlights the damaged chromocenter. The scatter plots represent the area of the damaged chromocenters normalized to the same chromocenters before UVC laser. **b** Procedure for targeting GFP-tagged DDB2 to major satellites sequences in pericentric heterochromatin. Confocal sections showing the aspect of pericentric heterochromatin domains (HC) upon tethering of the indicated GFP-tagged proteins in NIH/3T3 (**c**) or NIH/3T3 GFP-DDB2 cells (**d**). Heterochromatin tethering is relieved by expressing an anti-Cas9 peptide (**d**). The scatter plots show changes in volume and sphericity of heterochromatin domains quantified on reconstructed 3D images. Data are presented as mean values ± SD from n cells scored in at least three independent experiments. Statistical significance is calculated via two-sided Student's *t* test with Welch's correction when necessary (**a**, **d**). Multiple comparisons in (**c**) are performed by one-way ANOVA with Bonferroni post-test. Scale bars, 10 μm. Zoomed in views of heterochromatin domains (×2.6). Source data are provided as a Source Data file.

observed such differential behaviour of linker and core histones both in damaged heterochromatin and euchromatin, pointing to a general chromatin response to UV damage. Noteworthy, the local depletion of H1 was only detectable in cells expressing hDDB2 and not in the parental mouse cell line (DDB2 deficient) (Fig. 3b). Together, these findings establish that DDB2 promotes the displacement of linker histones H1 from damaged chromatin regions, which likely contributes, at least in part, to chromatin decompaction following UV damage.

**UV damage repair operates within heterochromatin domains.** To evaluate how UV damage repair proceeds in heterochromatin domains, we first analyzed the kinetics of DDB2 recruitment. We

observed that DDB2 recruitment to UV damage was not delayed in heterochromatin compared to euchromatin (Fig. 4a). DDB2-mediated decompaction of damaged heterochromatin could facilitate access of downstream repair factors to the core of heterochromatin domains. We thus examined the recruitment to UVC-damaged heterochromatin of repair proteins acting downstream of DDB2 in the NER pathway, namely, the intermediate repair factor XPB, which contributes to opening the damaged DNA double-helix, and the late repair factor proliferating cell nuclear antigen (PCNA), involved in repair synthesis after damage excision (Fig. 4b). Similar to what observed for GFP-hDDB2, we detected the accumulation of endogenous XPB and PCNA in damaged heterochromatin upon cell exposure to local UVC irradiation (Fig. 4c). Importantly, we noticed that PCNA

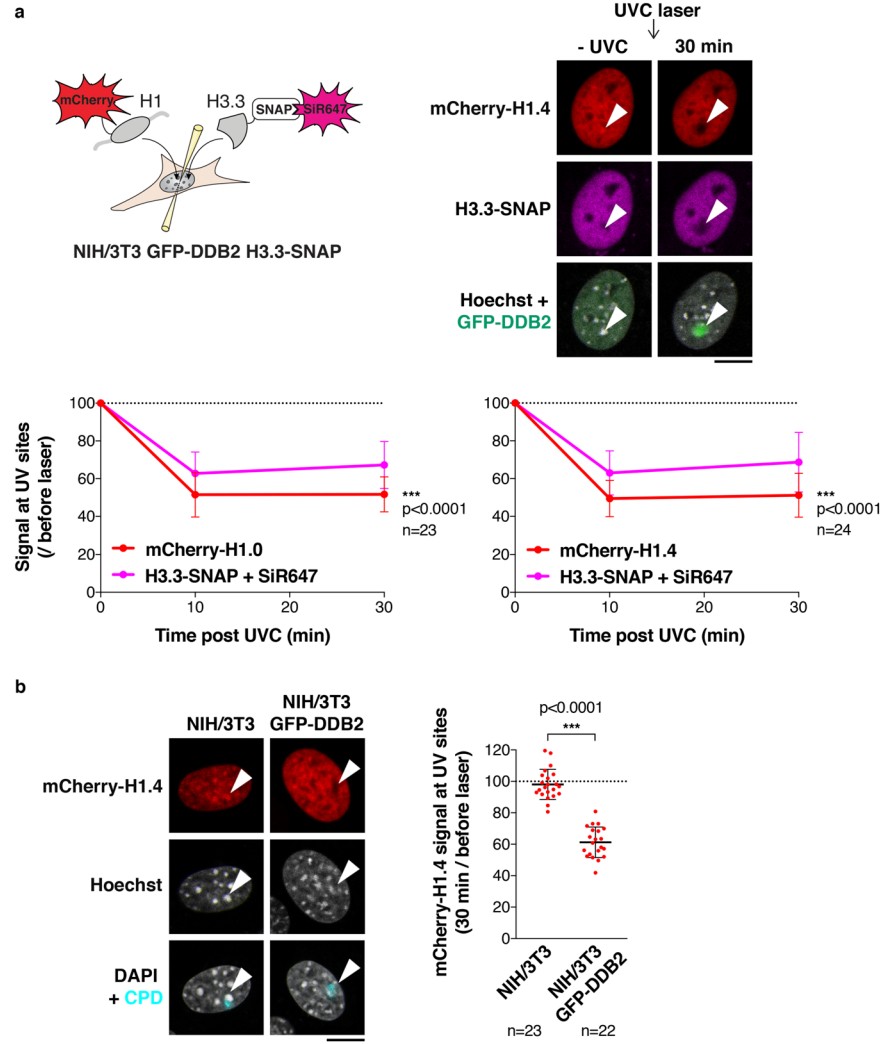

**Fig. 3 The UV damage sensor DDB2 promotes linker histone displacement from damaged chromatin. a** Scheme of the experiment for simultaneous detection of H1 and H3.3 in live cells exposed to UVC laser damage. H1 variants are transiently expressed as mCherry-tagged fusions in NIH/3T3 GFP-DDB2 cells stably expressing H3.3-SNAP, which is labelled with SNAP-cell SiR-647. The levels of H1 variants and H3.3 are measured in UVC-damaged regions, identified by GFP-DDB2 accumulation (white arrowheads), relative to the whole nucleus at the indicated time points after laser damage. Results normalized to before laser damage are presented on the graphs. **b** mCherry-H1.4 signal in damaged heterochromatin domains (white arrowheads) 30 min after UVC laser micro-irradiation analyzed by live imaging in the indicated cell lines. CPD staining in fixed cells highlights the damaged chromocenter. The scatter plot represents the mCherry-H1.4 signal loss in UVC-damaged chromatin regions in both cell lines. Data are presented as mean values ± SD from n cells scored in at least three independent experiments. Comparisons of histone signal loss are based on non-linear regression with a polynomial quadratic model (**a**). Statistical significance in (**b**) is calculated via two-sided Student's *t* test. Scale bars, 10 μm. Source data are provided as a Source Data file.

accumulated within heterochromatin domains during DNA damage repair, as observed both in our mouse cell line model and in human MCF7 cells that endogenously express DDB2 (Fig. 4d and Supplementary Fig. 5a). The recruitment of PCNA to the core of heterochromatin domains following UV damage contrasts with PCNA peripheral localization during heterochromatin replication[61]. This indicates that, unlike replicative synthesis, UV damage repair synthesis takes place inside heterochromatin domains. Altogether, these results establish that pericentromeric heterochromatin is fully permissive for NER factor recruitment up to late repair steps.

**Repair-coupled deposition of new H3 histones in heterochromatin domains.** UV damage repair elicits the deposition of newly synthesized histones in human cells[50,62–64], including the H3 histone variants H3.1 and H3.3. To investigate whether

such repair-coupled histone deposition was taking place in damaged heterochromatin, we examined the recruitment of H3 variant-specific chaperones, starting with the H3.1 histone chaperone chromatin assembly factor-1 (CAF-1), which is known to interact with PCNA during repair[65] and to deposit new H3.1 histones at UVC damage sites[64]. Similar to PCNA (Fig. 4c, d), we observed that CAF-1 accumulated in damaged heterochromatin upon local UVC irradiation both in mouse NIH/3T3 GFP-hDDB2 cells and in human MCF7 cells (Fig. 5a and Supplementary Fig. 5b). Regarding H3.3 histone chaperones, both HIRA (Histone Regulator A) and DAXX (Death Domain Associated Protein) can drive H3.3 deposition (reviewed in[66]). HIRA deposits H3.3 within transcribed euchromatin in mammalian cells[67,68] and in UVC-damaged chromatin in human cells[50], while DAXX promotes H3.3 enrichment at repeated sequences including pericentric heterochromatin[67,69,70]. We thus examined whether one or

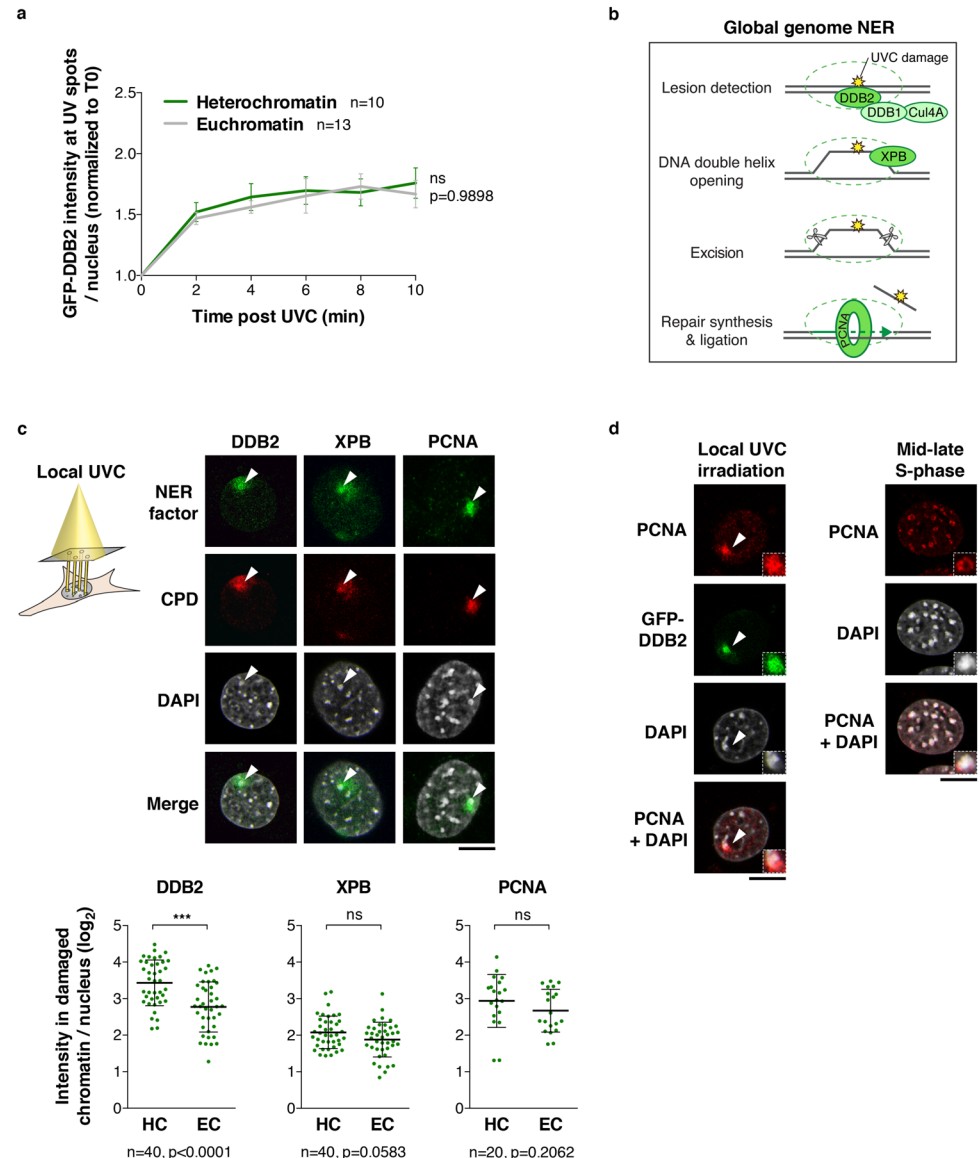

**Fig. 4 The NER pathway operates within heterochromatin domains. a** Kinetics of DDB2 recruitment to damaged euchromatin and heterochromatin analyzed in NIH/3T3 GFP-DDB2 cells after UVC laser damage. **b** Scheme of the Global Genome Nucleotide Excision Repair factors studied. **c** Recruitment to UVC damage (CPD) of early (DDB2), intermediate (XPB) and late (PCNA) repair factors, analyzed by immunofluorescence 30 min after local UVC irradiation through micropore filters in NIH/3T3 GFP-DDB2 cells. Cells with damaged heterochromatin domains (white arrowheads) were selected for the analysis. PCNA accumulation to damaged heterochromatin was analyzed outside S-phase. XPB and PCNA were not stained in green because the cells express GFP-DDB2, but are presented in green for simplicity. Scatter plots represent log2 fold enrichments of repair proteins in damaged heterochromatin (HC) and damaged euchromatin (EC) compared to the whole nucleus. **d** Accumulation of PCNA within heterochromatin domains upon local UVC irradiation and confined to the periphery of replicating heterochromatin in mid-late S-phase. Similar results were obtained in two independent experiments. Data are presented as mean values ± SD from n cells scored in at least two independent experiments. Comparisons of GFP-DDB2 kinetics in distinct chromatin domains are based on non-linear regression with a polynomial quadratic model (**a**). Statistical significance in (**c**) is calculated via two-sided Student's *t* test. All microscopy images are confocal sections. Scale bars, 10 µm. Insets show zoomed in views of heterochromatin domains (×2.3). Source data are provided as a Source Data file.

both of these chaperones were recruited to UVC-damaged heterochromatin. We observed that while HIRA accumulated in a comparable manner in damaged euchromatin and heterochromatin domains (Fig. 5b), DAXX was specifically recruited to damaged heterochromatin (Fig. 5c). Thus, although originally considered to operate in distinct chromatin domains[71], HIRA and DAXX chaperones co-exist within damaged heterochromatin, which we confirmed in human MCF7 cells (Supplementary Fig. 5c). In light of these findings, we explored the possibility of a co-recruitment of these two chaperones to damaged heterochromatin. However, siRNA-mediated depletion of HIRA did not impair DAXX recruitment and reciprocally, showing that both H3.3 histone chaperones are independently recruited to UVC-damaged heterochromatin (Supplementary Fig. 6a).

DAXX recruitment being heterochromatin specific, we further investigated the underlying mechanisms. We noticed a co-enrichment on damaged heterochromatin of the DAXX-binding partner and the heterochromatin-associated protein ATRX (Alpha Thalassaemia/Mental Retardation Syndrome X-Linked)

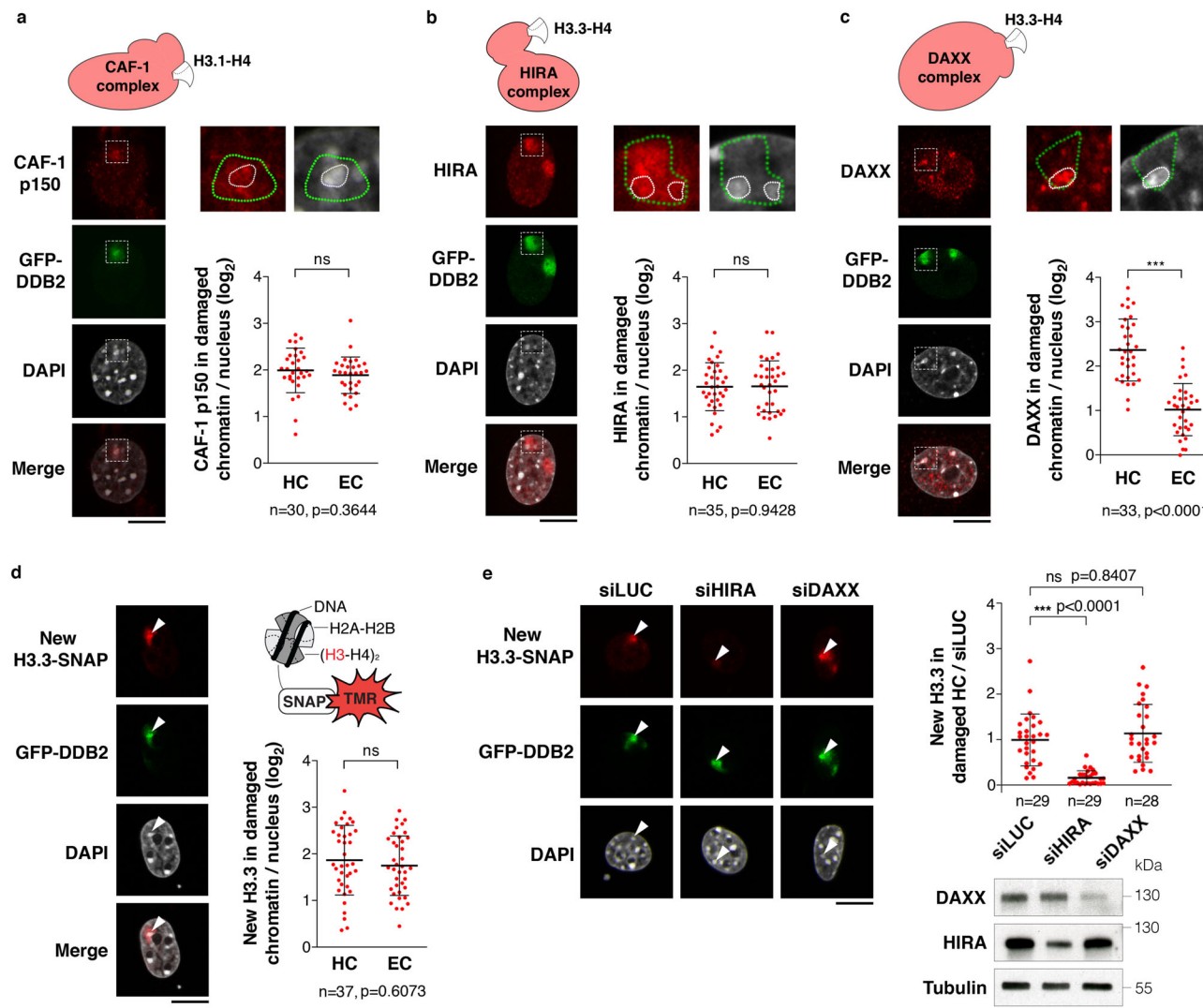

**Fig. 5 Histone H3 deposition in UVC-damaged heterochromatin.** Recruitment of the H3.1 histone chaperone CAF-1 (p150 subunit) (**a**), and of the H3.3 histone chaperones HIRA (**b**) and DAXX (**c**) to UVC-damaged regions, analyzed by immunofluorescence in NIH/3T3 GFP-DDB2 cells 1h30 after local UVC irradiation through micropore filters. Zoomed in views (×2.6) show damaged regions (delineated by green dotted lines) containing heterochromatin domains (delineated by white dotted lines). **d, e** Accumulation of newly synthesized H3.3 histones in UVC-damaged heterochromatin regions (white arrowheads) analyzed in NIH/3T3 GFP-DDB2 H3.3-SNAP cells 45 min after local UVC irradiation through micropore filters. H3.3 chaperones are knocked down by siRNA (siLUC, control) (**e**). siRNA efficiencies are controlled by western blot (Tubulin, loading control). Scatter plots represent log2 fold enrichments of histone chaperones or new H3.3-SNAP histones in damaged heterochromatin (HC) and damaged euchromatin (EC) compared to the whole nucleus (**a–d**) or normalized to the corresponding siLUC experiment (**e**). Data are presented as mean values ± SD from n cells scored in at least three independent experiments. Statistical significance is calculated via two-sided Student's t test with Welch's correction when necessary (**a–d**). Multiple comparisons in (**e**) are performed by one-way ANOVA with Bonferroni post-test. All microscopy images are confocal sections. Scale bars, 10 μm. Source data are provided as a Source Data file.

(Supplementary Fig. 6b). ATRX knockdown revealed that ATRX was driving DAXX recruitment to damaged heterochromatin (Supplementary Fig. 6c). Noteworthy, we observed DAXX accumulation in damaged heterochromatin both in and outside S-phase (Supplementary Fig. 6d), ruling out the possibility that DAXX recruitment could be coupled to heterochromatin replication, owing to enhanced chromatin accessibility during this process. Analogous to what observed for HIRA recruitment to UV sites in human cells[50], DAXX accumulation in UVC-damaged heterochromatin domains was dependent on the UV damage sensor DDB2 (Supplementary Fig. 6e).

We next explored the functional relevance of HIRA and DAXX recruitment to damaged heterochromatin. We first tested their possible contribution to repair of heterochromatin damage by analyzing GFP-DDB2 removal from UV-damaged chromocenters,

as a readout of repair progression (Supplementary Fig. 6f). These experiments ruled out a function for any of these chaperones in UV damage repair in heterochromatin domains. However, in line with the recruitment of H3.3-specific histone chaperones, we observed an accumulation of newly synthesized H3.3-SNAP histones within damaged heterochromatin domains, comparable to neighbouring euchromatin regions (Fig. 5d). Loss-of-function experiments revealed that only HIRA depletion markedly inhibited new H3.3 deposition in UV-damaged chromocenters (Fig. 5e). Although we cannot exclude a minor contribution of DAXX, HIRA thus appears to be the main driver for new H3.3 deposition in damaged heterochromatin. Collectively, these findings demonstrate that UVC damage drives the recruitment of H3 histone chaperones and new H3 deposition in hetero-chromatin domains.

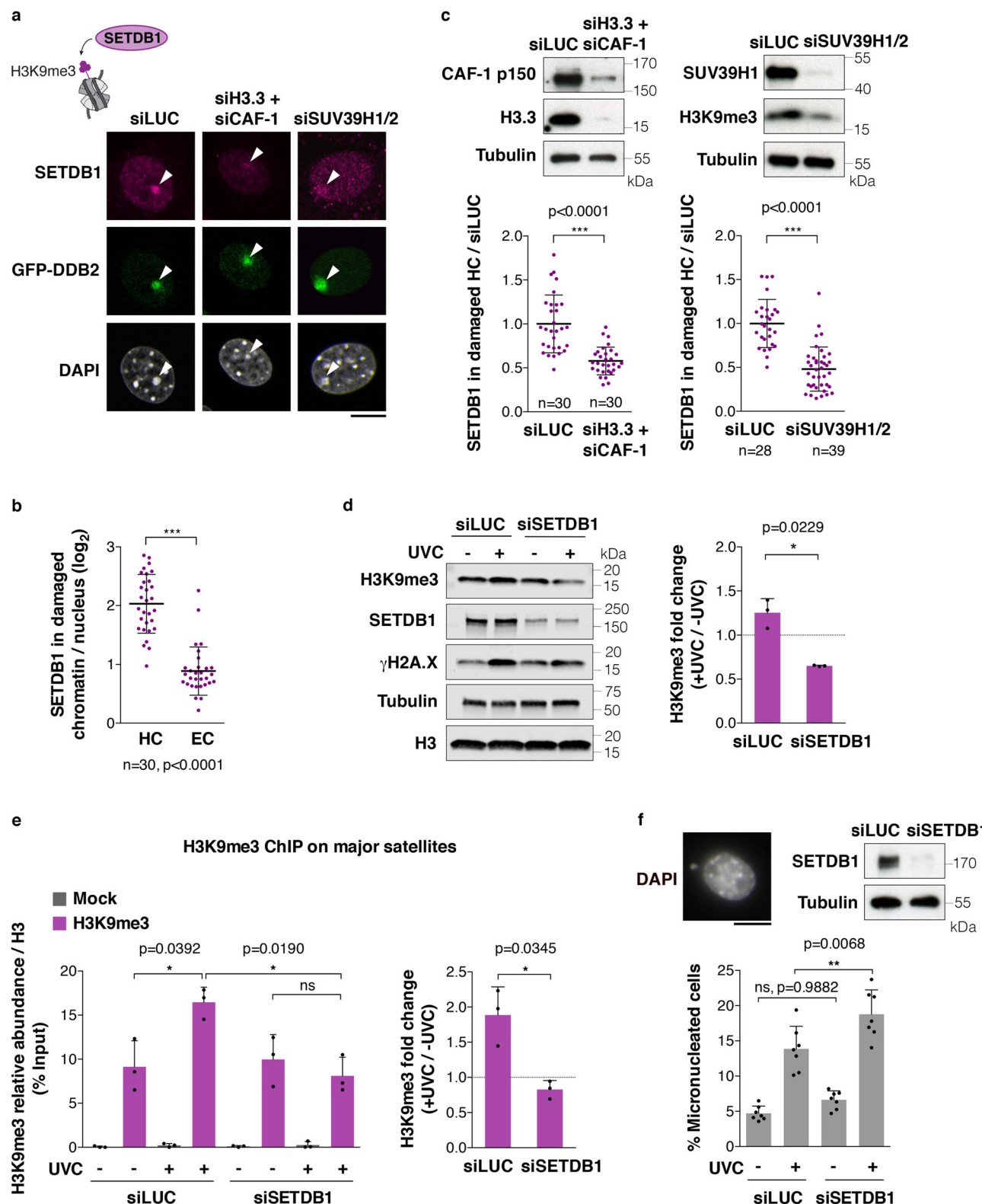

**SET domain bifurcated 1 (SETDB1) coordinates new H3 histone deposition and H3K9me3 maintenance in damaged heterochromatin**. Newly synthesized H3 histones do not carry the same post-translational modifications as nucleosomal H3 and are largely devoid of trimethylation marks[72,73]. Thus, we wondered whether and how the newly synthesized H3 histones deposited in damaged heterochromatin would acquire heterochromatin-specific modifications, including H3K9me3,

which we observed was maintained after UVC damage (Fig. 1c). Interestingly, when we examined the recruitment of H3K9 trimethyltransferases, we found that SETDB1 was specifically recruited to damaged pericentric heterochromatin (Fig. 6a, b) in a DDB2-dependent manner (Supplementary Fig. 7a), while SUV39H1 displayed only a slight enrichment on UVC-damaged compared to undamaged chromocenters, independently of SETDB1 (Supplementary Fig. 7b). The H3K27

**Fig. 6 SETDB1 is recruited to UVC-damaged heterochromatin and promotes H3K9me3 maintenance. a–c** Recruitment of SETDB1 to damaged heterochromatin (white arrowheads) in cells treated with the indicated siRNAs (siLUC, control) analyzed by immunofluorescence 1h30 after local UVC irradiation through micropore filters in NIH/3T3 GFP-DDB2 cells. Similar results were obtained in three independent experiments. **d** Changes in total H3K9me3 levels 1h30 post global UVC irradiation, detected by western blot on total extracts from NIH/3T3 GFP-DDB2 cells treated with the indicated siRNAs (siLUC, control). Tubulin, loading control; γH2A.X, damage marker. **e** H3K9me3 abundance on major satellites detected by ChIP before and 1h30 after global UVC irradiation in NIH/3T3 GFP-DDB2 cells treated with the indicated siRNAs (siLUC, control). Mock, no antibody. **f** Percentage of micronucleated cells before and 24 h after global UVC irradiation of NIH/3T3 GFP-DDB2 cells treated with the indicated siRNAs (siLUC, control). Scatter plots represent log2 fold enrichments of SETDB1 in damaged heterochromatin (HC) and damaged euchromatin (EC) compared to the whole nucleus (**b**) or normalized to the corresponding siLUC experiment (**c**). Bar graphs represent H3K9me3 abundance in damaged/undamaged conditions ((**d**, **e**) right) or H3K9me3/total H3 levels normalized to the input ((**e**) left). Data are presented as mean values ± SD from three (**d**, **e**) and seven (**f**) experiments or from n cells scored in at least three independent experiments. Statistical significance is calculated via two-sided Student's *t* test with Welch's correction when necessary ((**b–e**) right). Multiple comparisons are performed by one-way ANOVA with Bonferroni post-test ((**e**) left, (**f**)). All microscopy images are confocal sections. Scale bars, 10 μm. Source data are provided as a Source Data file.

trimethyltransferase enhancer of zeste homologue 2, in contrast, did not show any significant accumulation in damaged chromocenters (Supplementary Fig. 7c).

Given the specific recruitment of SETDB1 to damaged heterochromatin, we further investigated its potential role in methylating newly deposited H3 histones following UV damage. SETDB1 was dispensable for UV damage repair in heterochromatin domains (Supplementary Fig. 7d) and for new H3.3 deposition (Supplementary Fig. 7e). However, abrogation of new H3 histone deposition, achieved by simultaneous depletion of H3.3 and of the H3.1-chaperone CAF-1, impaired SETDB1 recruitment to UVC-damaged heterochromatin (Fig. 6a, c). Single depletion of H3.3 or CAF-1 had no or a very modest effect (Supplementary Fig. 7f). Furthermore, erasing parental H3K9me3 by knocking down SUV39H1/2 prevented SETDB1 recruitment to damaged heterochromatin (Fig. 6a, c). These results indicate that SETDB1 recruitment to damaged heterochromatin domains is driven by the deposition of newly synthesized H3 histones and also by SUV39H1/2 enzymes, which maintain parental H3K9me3. An attractive possibility is thus that SETDB1 may trimethylate newly deposited H3 histones in UV-damaged heterochromatin by copying the K9me3 mark from neighbouring SUV39H1/2-modified parental histones.

To assess the possible role of SETDB1 in H3K9me3 maintenance following UV damage, we first analyzed H3K9me3 total levels post-UV by western blot (Fig. 6d). These experiments revealed that, contrary to control cells where H3K9me3 levels increase moderately following UV irradiation (Supplementary Fig. 3f and Fig. 6d), SETDB1-knocked down cells displayed a significant reduction in H3K9me3 levels post-UV (Fig. 6d). Similar results were obtained by focusing on pericentric heterochromatin, as observed by chromatin immunoprecipitation (ChIP) of H3K9me3 on major satellite repeats (Fig. 6e). These data establish that SETDB1 contributes to H3K9me3 maintenance in UV-damaged heterochromatin.

Defective maintenance of pericentric heterochromatin may have deleterious consequences on chromosome segregation resulting in mitotic defects. To further investigate the functional relevance of SETDB1-dependent maintenance of heterochromatin in response to UV damage, we scored micronucleated cells 24 h following UVC irradiation and found higher rates of micronuclei in SETDB1 knockdown conditions (Fig. 6f). These results illustrate the importance of heterochromatin maintenance by SETDB1 in protecting cells against genome instability.

Together, these findings demonstrate that de novo deposition of H3 histones in UV-damaged heterochromatin stimulates the recruitment of the histone-modifying enzyme SETDB1, which promotes H3K9me3 maintenance thus preserving genome and epigenome integrity.

## Discussion

By assessing the consequences of UVC damage on mammalian pericentric heterochromatin domains, we provide important novel insights into the mechanisms for heterochromatin maintenance following DNA damage. We describe damage-mediated alterations in heterochromatin compaction with the retention of silencing histone marks, which may facilitate repair in compact regions of the genome while preserving heterochromatin identity. We also unveil a repair-coupled deposition of newly synthesized histones in damaged heterochromatin, and propose that histone chaperones and chromatin modifiers cooperate to maintain heterochromatin integrity following DNA damage (Fig. 7).

**Regulation of heterochromatin compaction following UV damage.** Chromatin reorganization coupled to the early stages of the DNA damage response is considered to be critical for efficient DNA repair[17,19]. This is particularly relevant in compact heterochromatin domains. Indeed, decompaction of pericentromeric heterochromatin has been reported in response to radiation- and nuclease-induced breaks both in flies[43] and in mammalian cells[44,46], where it correlates with processive DNA synthesis[74], as well as following heat stress in plants[75]. Here, we provide evidence for pericentric heterochromatin decompaction following UVC damage in mouse fibroblasts. Mechanistically, the regulation of damaged heterochromatin compaction likely differs in response to distinct types of DNA damage because it involves damage-specific factors. Indeed, in response to UV lesions, we have identified the UV damage sensor DDB2 as a master regulator of heterochromatin compaction and we have shown that DDB2 promotes the displacement of linker histones H1 from damaged chromatin. Noteworthy, H1 is also displaced from chromatin at sites of DNA breaks but the underlying mechanisms are still to be characterized[24,27]. Recent structural data indicate that the DDB2 complex can expose UV lesions occluded in nucleosomal DNA by promoting DNA shifting[76]. Such local activity at the nucleosome level is unlikely to sustain larger scale chromatin decompaction but it could stimulate displacement of H1, which bridges DNA at the entry and exit sites of the nucleosome core particle. Further studies, including single molecule approaches[77], will be needed to fully dissect the molecular bases of DDB2-mediated H1 release. It will also be interesting to investigate whether DDB2 crosstalks with the histone chaperone SET, which evicts H1 from chromatin thus impacting cell survival following DNA damage[78]. Given the important role of linker histones in higher-order chromatin folding[57] it is tempting to speculate that DDB2 triggers damaged chromatin decompaction at least in part by promoting H1 displacement. Importantly, the function of DDB2 in regulating chromatin compaction likely extends to other types of DNA lesions than those processed by the NER pathway considering that

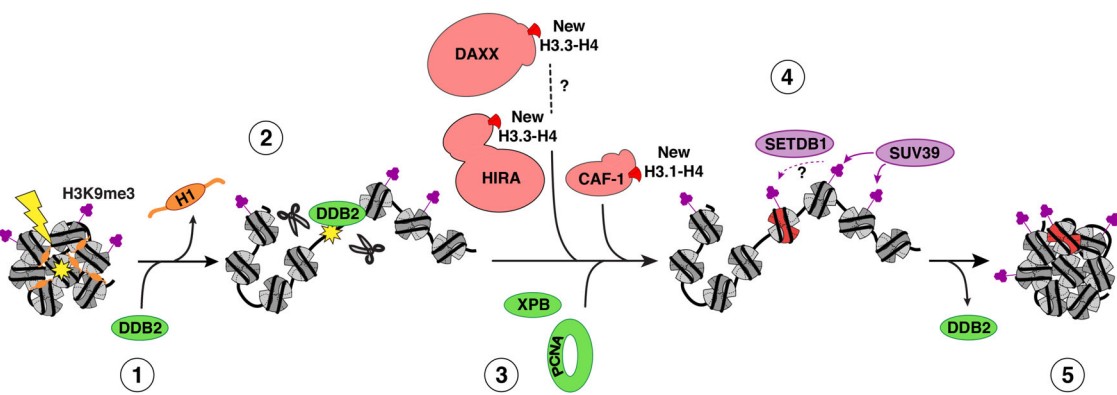

**Fig. 7 Model for heterochromatin maintenance following UVC damage.** Recognition of UVC damage by the sensor protein DDB2 (1) leads to linker histone H1 displacement and decompaction of damaged pericentric heterochromatin (2), thus facilitating access of downstream repair factors and histone chaperones (3) to the core of the domain. Histone chaperones promote the incorporation of newly synthesized H3 histones (in red), which subsequently acquire heterochromatin-specific modifications through the action of histone methyltransferases (4). DDB2 release during repair progression allows heterochromatin recompaction (5).

DDB2 also detects oxidative damage and contributes to base excision repair[79,80].

It will also be of major interest to assess the impact of damage-mediated chromatin decompaction on the three-dimensional organization of chromatin in the nuclear space[81]. Indeed, it is not known whether decompaction entails only local chromatin movement with the loss or enlargement of chromatin loops within topologically associated domains, or more profound and global alterations of chromatin topology.

Functionally, whether heterochromatin decompaction facilitates the access of repair factors to damaged DNA is not entirely clear. Here, we have shown that the NER machinery can access UV lesions in pericentric heterochromatin and that repair can be completed within these domains. This contrasts with the relocalization of repair foci to the periphery of heterochromatin domains for late steps of DSB recombination[44,46], as also observed for replication foci[61], thus showing that not all pathways that involve DNA synthesis are excluded from the core heterochromatin domain. Our results are consistent with several studies showing that, even if volume exclusion and moderate diffusive hindrance occur in heterochromatin domains[82], heterochromatin is accessible to large proteins[83], including non-homologous end joining, single-strand annealing and early homologous recombination (HR) factors[46]. Considering that NER, unlike HR of DSBs, does not pose a risk for ectopic recombination between heterochromatic repeats, there would be no need for a relocalization of the NER machinery to the heterochromatin periphery and thus no spatial segregation of UV damage repair events. Instead, there is a temporal regulation of NER in heterochromatin, with slower kinetics of UV damage repair[34,35], likely due to the necessary decompaction to promote access to lesions buried in heterochromatin.

**Histone deposition in UV-damaged heterochromatin: role of histone chaperones.** By assessing the recruitment of H3 variant-specific histone chaperones to UVC-damaged heterochromatin, we have identified the histone chaperone HIRA as the main driver of new H3.3 deposition at UVC-damaged heterochromatin. While we cannot formally exclude that the DAXX-ATRX complex has a minor contribution to this process, we can envision alternative roles for this complex, such as stimulating parental H3.3 recovery during the repair response. Known as a promiscuous histone chaperone, the DAXX-ATRX complex could also regulate the dynamics of other histone variants, including CENP-A[84] and macroH2A1[85,86]. Another potential

role would be the control of heterochromatin recompaction at late time points post-UV, in light of data revealing the importance of DAXX for the structural organization of pericentric heterochromatin in mouse cells[87] and of DAXX-ATRX-mediated deposition of H3.3 for chromocenter clustering during myogenic differentiation[88]. Finally, it has been proposed that this histone chaperone complex could regulate repair synthesis as observed during HR[89], but our preliminary observations do not support such a function during NER.

**Maintenance of silencing histone marks in UV-damaged heterochromatin.** We have observed that UV-damaged heterochromatin decompacts while retaining silencing histone marks. This is in line with previous studies in response to DSBs[46,90], and highlights an uncoupling between chromatin structural and molecular determinants during DNA damage repair. Reciprocally, heterochromatic histone marks can be erased without any significant effect on heterochromatin decompaction, as observed upon SUV39H1/2 loss of function (Figs. 1a and 6a)[53]. We propose that by retaining their histone marks, heterochromatin domains also maintain their identity, which could be crucial for the re-establishment of the original chromatin state once DNA repair is complete. In addition to heterochromatin histone marks, HP1α, which is recruited to UVC lesions[91], could contribute to preserve heterochromatin identity following DNA damage, at least in terms of transcription silencing because HP1α does not control heterochromatin compaction.

Mechanistically, we have established that the histone methyltransferase SETDB1 promotes H3K9me3 maintenance in UV-damaged heterochromatin. In line with these findings, SETDB1 has been involved in DNA damage-induced H3K9me3 leading to sex chromosome inactivation in meiosis[92]. We have found that the histone methyltransferase SETDB1 is specifically recruited to UVC-damaged heterochromatin. Although the underlying mechanisms are still unclear, they likely involve SETDB1 association with CAF-1[93] and SUV39 enzymes[94]. They may also implicate SETDB1 tandem Tudor domains, reported to bind specifically to dually modified histone H3 containing both K14 acetylation (H3K14ac) and K9 methylation (H3K9me1/2/3)[95]. Given that newly synthesized H3 histones are enriched in K14ac[96], and are not optimal substrates for trimethylation by SUV39H1/2[97], we hypothesize that SETDB1 could bind, via its Tudor domains, the new H3 histones deposited in damaged heterochromatin, and then trimethylate these histones, thus mirroring SUV39H1/2-dependent trimethylation on parental H3.

Future studies will determine whether SETDB1 indeed promotes trimethylation of newly deposited H3 histones in UV-damaged heterochromatin. While H3K9me3 is maintained, and even slightly increased, in UV-damaged heterochromatin, we did not find evidence for ectopic H3K9me3 formation in damaged euchromatic regions. This contrasts with the increase in H3K9me3 reported after DSB induction, which contributes to transcriptional repression[98,99]. While our study underlines the maintenance of heterochromatin marks upon genotoxic stress, loss of heterochromatin marks can also be key for the preservation of genome integrity as observed with the reduction of H3K9me3 heterochromatin upon mechanical stress, which protects mammalian cells against DNA damage[100].

Collectively, our work sheds new light on the processes safeguarding pericentric heterochromatin integrity following DNA damage. It would be of interest to determine if similar or distinct mechanisms operate in other heterochromatin domains characterized by different patterns of epigenetic marks, such as telomeric chromatin and facultative heterochromatin. Beyond the DNA damage response, our findings may also provide a molecular framework for understanding heterochromatin maintenance during other disruptive events in both normal and pathological conditions, like DNA replication, cell differentiation, aging and disease.

## Methods

**Cell culture and drug treatments.** U2OS (ATCC HTB-96, human osteosarcoma, female), MCF7 (ATCC HTB-22, human breast adenocarcinoma, female) and NIH/3T3 cells (ATCC CRL-1658, mouse embryonic fibroblast, male) were grown at 37 °C and 5% $CO_2$ in Dulbecco's modified Eagle's medium (Invitrogen) supplemented with 10% foetal bovine serum (EUROBIO) and antibiotics (100 U/ml penicillin and 100 μg/ml streptomycin, Invitrogen) and the appropriate selection antibiotics (Euromedex, Supplementary Table 1). For seeding NIH/3T3 cells on coverslips, coverslips were first coated with 20 μg/ml Collagen Type I (MERCK Millipore) and 2 μg/ml fibronectin (Sigma-Aldrich) to increase cell adhesion. The PARP inhibitor olaparib KU58948/AZD2281 (Selleckchem) was added to the culture medium at 10 μM final 1 h before UV damage.

**siRNA and plasmid transfections.** siRNA purchased from Eurofins MWG Operon (Supplementary Table 2) were transfected into cells using Lipofectamine RNAiMAX (Invitrogen) following manufacturer's instructions. The final concentration of siRNA in the culture medium was 50–80 nM. Cells were harvested 48–72 h after transfection.

Cells were transfected with plasmid DNA (Supplementary Table 3)[101–107] using Lipofectamine 2000 (Invitrogen) according to manufacturer's instructions. For stable cell line establishment (Supplementary Table 1), plasmid DNA was transfected into cells at 1 μg/ml final, 48 h before antibiotic selection of clones. For transient transfections, each plasmid was at 0.5 μg/ml final and cells were fixed 48 h post transfection. For DDB2 tethering to major satellites, plasmids encoding GFP-tagged proteins, GBP-dCas9-mRFP and major satellite gRNA were co-transfected into NIH/3T3 cells 48 h before cell analysis. For DDB2 detachment from major satellites, NIH/3T3 GFP-DDB2 cells were transfected with GBP-dCas9-mRFP and major satellite gRNA plasmids and 24 h later with anti-Cas9 plasmid. Cells were fixed 24 h after the second transfection.

**UVC irradiation.** Cells grown on glass coverslips (12 mm diameter, thickness No.1.5, Thorlabs) were irradiated with UVC (254 nm) using a low-pressure mercury lamp. Conditions were set using a VLX-3W dosimeter (Vilbert-Lourmat). For global UVC irradiation, cells in phosphate buffer saline (PBS) were exposed to UVC doses ranging from 4 to 12 J/m$^2$ for survival assays and to 10 J/m$^2$ in other experiments. For local UVC irradiation[108,109], cells were covered with a polycarbonate filter (5 μm pore size, Millipore) and irradiated with 150 or 300 J/m$^2$ UVC. Irradiated cells were allowed to recover in culture medium for the indicated times before fixation.

For UVC laser micro-irradiation[110], cells were grown on quartz coverslips (25 mm diameter, thickness No.1, SPI supplies) and nuclei were stained by adding Hoechst 33258 (10 μg/mL final, Sigma-Aldrich) to the culture medium 30 min before UVC irradiation. Quartz coverslips were transferred to a Chamlide magnetic chamber on a custom stage insert (Live Cell Instrument) and cells were irradiated for 50 ms using a 2 mW pulsed diode-pumped solid-state laser emitting at 266 nm (RappOptoElectronics, Hamburg GmbH) directly connected to a Zeiss LSM700 confocal microscope adapted for UVC transmission with all-quartz optics. The laser was attenuated using a neutral density filter OD1 and focused through a ×40/0.6 Ultrafluar glycerol objective with quartz lenses. By comparing UVC laser-

induced damage to damage induced by the UVC lamp based on CPD staining (Supplementary Fig. 2d), the UVC dose delivered at the site of laser micro-irradiation in our murine cellular model can be estimated at 600 J/m$^2$, which generates on average 1 pyrimidine dimer every nucleosome (1 UV lesion/ 150 bp) in 2% of the nuclear volume.

**UVA laser damage.** Cells were incubated with Hoechst 33258 (20 μg/mL final, Sigma-Aldrich) 30 min before laser damage. Damage was introduced with a 405 nm laser diode (3 mW) focused through a Plan-Apochromat ×63/1.4 oil objective on a LSM780 confocal microscope (Zeiss) using the following laser settings: 15% power, 10 iterations, scan speed 12.6 μs/pixel.

**Cell extracts and western blot.** Total extracts were obtained by scraping cells on plates or resuspending cell pellets in Laemmli buffer (50 mM Tris-HCl pH 6.8, 1.6% sodium dodecyl sulfate (SDS), 8% glycerol, 4% β-mercaptoethanol, 0.0025% bromophenol blue) followed by 5 min denaturation at 95 °C. Alternatively, cell pellets were resuspended in lysis buffer (1 M Tris-HCl pH 6.8, 50 mM NaCl, 0.5% NP-40, 1%, Sodium Deoxycholate, 1% SDS, 5 mM MgCl2) before addition of Laemmli buffer with 0.25 U/μL benzonase (final concentration, Merck Millipore) for 10 min followed by 5 min denaturation at 95 °C.

For western blot analysis, extracts were run on 4–20% Mini-PROTEAN TGX gels (Bio-Rad) in running buffer (200 mM glycine, 25 mM Tris, 0.1% SDS). Proteins were transferred onto nitrocellulose membranes (Amersham Protran) for 30 min at 15 V with a Trans-Blot SD semidry transfer cell (Bio-Rad) or in transfer buffer (25 mM Tris, 200 mM glycine, 20% ethanol) for 2 h at 52 V with a liquid transfer system (Bio-Rad). Total proteins were revealed by Pierce® Reversible Stain (Thermo Scientific). Proteins of interest were probed using the appropriate primary and horseradish peroxidase-conjugated secondary antibodies (Supplementary Table 4), detected using SuperSignal West Pico or Femto chemiluminescence substrates (Pierce) on hyperfilms MP (Amersham). When fluorescence detection was used instead of chemiluminescence, secondary antibodies were conjugated to IRDye 680RD or 800CW (Supplementary Table 4), membranes were scanned with an Odyssey Fc-imager (LI-COR Biosciences) and analyzed with Image Studio Lite software using total protein stain for normalization.

**GFP pull-down.** For GFP pull-downs[111], NIH/3T3 cells expressing GFP or GFP-DDB2 were collected in PBS (2–5 million cells per pull-down). Cell pellets were resuspended in 1 ml ice-cold lysis buffer (50 mM Tris pH 7.5, 250 mM NaCl, 0.5% NP-40, 2.5 mM MgCl$_2$, Roche EDTA-free Protease inhibitor cocktail) supplemented with benzonase (0.25 U/μL final, Sigma-Aldrich) and samples were incubated for 2 h at 4 °C under constant mixing. The cell lysates were cleared by centrifugation at full speed for 10 min at 4 °C and incubated overnight at 4 °C with 20 μl of GFP-Trap-A beads (Chromotek) in lysis buffer. Input samples were collected from cell lysates before the addition of beads. After five washes in lysis buffer, beads were boiled for 10 min at 95 °C in Laemmli buffer. Pull-down and input samples were analyzed by western blot.

**Flow cytometry.** For cell cycle analysis, cells were fixed in ice-cold 70% ethanol before DNA staining with 50 μg/ml propidium iodide (Sigma-Aldrich) in PBS containing 0.05% Tween and 0.5 mg/ml RNase A (USB/Affymetrix). DNA content was analyzed by flow cytometry (20,000 cells per condition) using a BD FACS-calibur flow cytometer (BD Biosciences) and FlowJo software (TreeStar).

**SNAP-tag labelling of histones.** For specific labelling of newly synthesized histones[52,112], cells were grown on glass coverslips and pre-existing SNAP-tagged histones were first quenched by incubating cells with 10 μM of the non-fluorescent substrate SNAP-Cell Block (New England Biolabs) for 30 min followed by a 30-min wash in fresh medium and a 2-h chase. The new SNAP-tagged histones synthesized during the chase were fluorescently labelled with 2 μM of the red-fluorescent reagent SNAP-cell TMR star or SiR-647 (New England Biolabs) during a 15-min pulse step followed by 30-min wash in fresh medium. Cells were subsequently permeabilized with Triton X-100, fixed and processed for immunostaining. Cells were irradiated with a UVC lamp before the pulse step. Labelling of total histones was performed by a 30-min pulse with 2 μM of the SNAP-cell SiR-647 reagent (New England Biolabs) followed by 30 min wash in fresh medium.

**Ethynyl-deoxyUridine (EdU)-labelling of replicating cells and repair sites.** To visualize replication foci, 10 μM EdU was incorporated into cells on glass coverslips during 15 min at 37 °C and revealed using the Click-It EdU Alexa Fluor 647 or 594 Imaging kit (Invitrogen) according to manufacturer's instructions. To localize the sites of UV damage repair, cells were incubated with 10 μM EdU for 1h30 after local UVC irradiation and EdU was revealed using the Click-It EdU Alexa Fluor 488 Imaging kit (Invitrogen).

**Nascent RNA labelling.** Cells on glass coverslips were incubated in medium supplemented with 0.5 mM EU for 45 min at 37 °C, and EU incorporation was revealed with Click-iT RNA Alexa Fluor 594 Imaging kit (Invitrogen) according to manufacturer's instructions. Coverslips were mounted in Vectashield medium with

DAPI (Vector laboratories). EU fluorescence intensity in heterochromatin was measured using ImageJ software. Heterochromatin segmentation was based on DAPI staining.

**Immunofluorescence**. Cells grown on coverslips were either fixed directly with 2% paraformaldehyde (Electron Microscopy Sciences) for 20 min and permeabilized for 5 min with 0.5% Triton X-100 in PBS or cells were pre-extracted before fixation with 0.5% Triton X-100 in CSK buffer (Cytoskeletal buffer: 10 mM PIPES pH 7.0, 100 mM NaCl, 300 mM sucrose, 3 mM MgCl$_2$) for 5 min at room temperature to remove soluble proteins. For PCNA staining, cells were fixed with 100% ice-cold methanol for 15 min. For the detection of UVC photoproducts (CPD), DNA was denatured with 2 N HCl for 10 min at 37 °C (Cosmo Bio antibody, Supplementary Table 4) or with 0.5 M NaOH for 5 min at room temperature (Kamiya antibody, Supplementary Table 4). Since this denaturation quenches GFP fluorescence, when CPD detection was combined with the visualization of GFP-DDB2, immuno-fluorescence was performed in two steps starting with GFP immunodetection using a rat anti-GFP antibody (Supplementary Table 4) followed by fixation, denatura-tion and CPD immunodetection. Samples were blocked for 10 min in 5% BSA (Bovine Serum Albumin, Sigma-Aldrich) in PBT (PBS 0.5% Tween-20), followed by 45 min incubation with primary antibodies and 30 min incubation with sec-ondary antibodies coupled to Alexa Fluor dyes (Supplementary Table 4) diluted in blocking buffer. Coverslips were mounted in Vectashield medium with DAPI (Vector laboratories).

**DNA-fluorescence in situ hybridization (DNA-FISH) of mouse major satel-lites**. Cells on quartz coverslips were fixed in 2% paraformaldehyde (Electron Microscopy Sciences). 5′TYE563-labelled locked nucleic acid probes (Exiqon) against mouse major satellite sequences (Supplementary Table 5) were precipitated with mouse Cot-1 DNA (Invitrogen) and Salmon Sperm DNA (Thermo Fisher Scientific), resuspended in formamide and denatured for 7 min at 75 °C. The probes were then diluted in an equal volume of 2X Hybridization Buffer (4X SSC, 20% Dextran Sulfate, 2 mg/ml BSA). Coverslips were dehydrated in 80, 90 and 100% ethanol and equilibrated in 2X SCC at 80 °C. Coverslips were then denatured for 10 min at 80 °C in 70% formamide/2X SSC (pH = 7.2), dehydrated in 70, 80, 90 and 100% ethanol and incubated overnight with the major satellite probes at 37 °C. After three 5-min washes in 50% formamide/2X SSC at 45 °C, and three 4-min washes in 2X SSC at 45 °C, coverslips were mounted in Vectashield medium with DAPI (Vector laboratories). Since sample denaturation quenches GFP fluores-cence, immunofluorescence against GFP (to detect GFP-DDB2) was performed prior to DNA-FISH.

**Image acquisition and analysis**. Fluorescence imaging was performed with a Leica DMI6000 epifluorescence microscope using a Plan-Apochromat ×40/1.3 or ×63/1.4 oil objective. Images were captured using a CCD camera (Photometrics) and Metamorph software. Images were assembled with Adobe Photoshop. For confocal imaging, samples were observed on Zeiss LSM710/780/980 confocal microscopes using Plan-Apochromat ×63/1.4 and ×40/1.3 oil objectives. Live-cell imaging coupled to UVC laser micro-irradiation was performed using a ×40/0.6 Ultrafluar Glycerol objective on a Zeiss LSM700 confocal microscope. Images were captured using Zen software, and analyzed with ImageJ (US National Institutes of Health, Bethesda, Maryland, USA, http://imagej.nih.gov/ij/). Nuclei and heterochromatin domains were segmented on 2D confocal images based on DAPI or Hoechst staining, and UVC-damaged regions based on GFP-DDB2 fluorescence using custom-made ImageJ macros. To measure histone density loss in damaged regions over time, a circular area was defined inside the damaged region based on GFP-DDB2 signal 1 min post laser damage and was kept the same for all time points. The fluorescence level of tagged-histone proteins in this damaged area was normalized to the fluorescence intensity in the entire nucleus at the same time point to correct for overall bleaching of the signal due to repetitive imaging and all results were normalized to before damage. The volume, sphericity and GFP intensity of heterochromatin domains as well as H3K9me3 intensity in damaged heterochromatin were analyzed on 3D images reconstructed from z-stacks using Imaris (Bitplane, http://www.bitplane.com/imaris).

**Colony forming assays**. Cells were replated 48 h after siRNA transfection and exposed to global UVC irradiation (4, 8 and 12 J/m$^2$) the following day. Colonies were stained 12 days later with 0.5% crystal violet/20% ethanol and counted. Results were normalized to plating efficiencies.

**Reverse transcription-quantitative polymerase chain reaction (RT-qPCR)**. Total RNA was extracted from cells with TRIzol™ reagent following manufacturer's instructions (Invitrogen) and precipitated in isopropanol. RNA samples were subject to DNA digestion with Turbo DNA-free (Invitrogen) before reverse tran-scription with Superscript III RT using random primers (200 ng/reaction, Invi-trogen). Quantitative PCR reactions were carried out with the indicated primer pairs (Eurofins MWG Operon, 500 nM final concentration, Supplementary Table 5) and Power SYBR® Green PCR Master Mix (Applied Biosystems) and read in MicroAmp® Fast Optical 96-well plates (Applied Biosystems) using an ABI 7500

Fast detection system (Applied Biosystems). Results were normalized to the amount of the GAPDH housekeeping gene product.

**Chromatin immunoprecipitation (ChIP)**. For ChIP experiments[113], cells were crosslinked for 15 min with 1% formaldehyde (Sigma-F8775). The fixation reaction was stopped by adding glycine (0.125 M final concentration) for 5 min. Cells were collected and resuspended in cell lysis buffer (5 mM Pipes, 85 mM KCl, 0.5% NP-40). Lysates were homogenized with a tight Dounce homogenizer (DWK Life Sciences) and nuclei were collected by centrifugation. Samples were resuspended in nucleus lysis buffer (50 mM Tris pH 8.1, 10 mM EDTA, 1% SDS) and sonicated using the Bioruptor plus water bath system (Diagenode, 16 cycles at high power, 30 s ON/30 s OFF) to an average fragment size of 0.5–1 kb as assessed by agarose gel electrophoresis. Chromatin was diluted in dilution buffer (0.01% SDS, 1.1% Triton X-100, 1.2 mM EDTA, 16.7 mM Tris pH 8.1, 167 mM NaCl) and 25 µg of chromatin was incubated overnight at 4 °C with primary antibodies (Supplemen-tary Table 4) or without antibody as negative control. Immune complexes were recovered with 20 µL blocked protein A/G magnetic beads (Thermo Scientific) during 4 h at 4 °C. Beads were washed once in dialysis buffer (2 mM EDTA, 50 mM Tris pH 8.1, 0.2% Sarkosyl) and four times in wash buffer (100 mM Tris pH 8.1, 500 mM LiCl, 1% NP-40, 1% Sodium Deoxycholate). Samples were resuspended in TE buffer (10 mM Tris pH 8.1, 1 mM EDTA pH 8.1), treated with RNase A and DNA was eluted from the beads by adding SDS (1% final concentration) overnight with constant mixing at 60 °C. After proteinase K treatment, DNA was purified by phenol:chloroform:isoamyl alcohol extraction (Invitrogen) and resuspended in TE buffer.

Quantitative PCR reactions were performed as described in the RT-qPCR section. All experiments included a standard curve and all samples were analyzed in triplicates. Results were normalized to the input.

**Micronuclei**. To assess the proportion of micronucleated cells, cells were seeded on glass coverslips and transfected with siRNAs. After 72 h, cells were irradiated globally with UVC (10 J/m$^2$) and allowed to recover in culture medium for 24 h before fixation. Coverslips were mounted in Vectashield medium with DAPI (Vector laboratories). At least 500 cells were counted for each condition.

**Statistical analysis**. Percentages of positively stained cells were obtained by scoring at least 150 cells in each experiment. Statistical tests were performed using GraphPad Prism. H3K9me3 relative abundance ± UV was compared to a theore-tical mean of 1 by one-sample $t$-test. $p$ values for mean comparisons between two groups were calculated with a two-sided Student's $t$ test with Welch's correction when necessary. Multiple comparisons were performed by one-way ANOVA with Bonferroni post-test. Comparisons of clonogenic survival were based on non-linear regression with a polynomial quadratic model. ns: non-significant, *$p < 0.05$, **$p < 0.01$, ***$p < 0.001$. Confidence interval of all statistical tests: 95%.

**Reporting summary**. Further information on research design is available in the Nature Research Reporting Summary linked to this article.

## Data availability

All data generated during this study are included in this article and its supplementary information files. Source data are provided with this paper and are available on Mendeley https://doi.org/10.17632/9pn8wbhr9x.1.

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

## Acknowledgements

We thank the members of our laboratory for stimulating discussions and P.-A. Defossez and C. Rougeulle for critical reading of the paper. We thank J. Bondy-Denomy, S. Bultmann, D. Lleres, N. Mailand and R. Nishi for sharing plasmids. We acknowledge the ImagoSeine facility (Institut Jacques Monod, France BioImaging) for confocal microscopy and the imaging platform of the Epigenetics and Cell Fate Centre for epifluorescence microscopy. This work was supported by the European Research Council (ERC starting grant ERC-2013-StG-336427 "EpIn" and consolidator grant ERC-2018-CoG-818625 "REMIND"), the French National Research Agency (ANR-12-JSV6-0002-01 and ANR-18-CE12-0017-01), the "Who am I?" laboratory of excellence (ANR-11-LABX-0071) funded by the French Government through its "Investments for the Future" programme (ANR-11-IDEX-0005-01), EDF Radiobiology programme RB 2014-01, the Fondation ARC and France BioImaging (ANR-10-INBS-04). S.E.P. is an EMBO Young Investigator. A.F. received a Ph.D. fellowship from the University of Paris and P.C. was supported by the Fondation ARC (PDF20190509195) and the Fondation Recherche Medicale (ARF201909009206).

## Author contributions

A.F., A.C., P.C. and S.E.P. designed and performed experiments, analyzed the data and wrote the paper. O.C. provided technical assistance and established mouse stable cell lines. O.L. and O.R. implemented the UVC laser technology and helped with image analyses. S.E.P. supervised the project.

## Competing interests

The authors declare no competing interests.
