## [Peer Review File · Nature Communications]

REVIEWER COMMENTS

Reviewer #1 (Remarks to the Author):

The authors have investigated the events occurring at heterochromatin following UV damage. In particular they found that damaged heterochromatin undergoes decompaction and then recompaction. Histone modifications and transcriptional silencing are maintained. While DPB2 influences heterochromatin compaction, SETDB1 is required to maintain histone marks following UV damage.

General comments

They used immunofluorescent techniques in combination with UVC laser coupled to confocal microscope to follow the events at the heterochromatin in engineered NIH/3T3 mouse embryonic fibroblasts. The experimental system is brave and original. Overall the paper is interesting and the findings are potentially relevant. However, there are crucial points that must be addressed.

Specific comments

1. The first chapter of the results section should describe the experimental system. This is important as the entire paper is based on this approach. In particular the levels of GFP-hDDB2 must be shown.
2. Is the decompaction/compaction influenced by NER processing/repair? In particular is the decompaction dependent on NER-mediated DNA processing? This is an important control to rule out other effects caused by laser treatment (such as protein/membrane damage).
3. Does the cell cycle influence heterochromatin decompaction? In particular it would be important to see G1 cells.

Reviewer #2 (Remarks to the Author):

Fortuny et al NCB Review

In their manuscript, "Imaging the response to DNA damage in heterochromatin domains reveals core principles of heterochromatin maintenance", Fortuny et al. explore the molecular mechanism of ultraviolet C light induced DNA damage repair within regions of constitutive heterochromatin using a targeted damage approach in an NIH3T3 murine and MCF7 human cancer-derived cell model. I found this study very well written, and am broadly supportive of the work as experiments are high quality, the topic is of broad interest, and they are asking mechanistic question. I do, however, have numerous concerns relating to the methodology that require clarification and/or added controls (see below).

General comment on all the data that lead to the final model shown in Figure 7. The authors present data for each step in a process for heterochromatin relaxation and recompacting during UV DNA damage responses. This is good, but I note that the details of each step in the mechanism (especially intermediate steps, i.e. the data in Figure 3-4) aren't really studied with the degree of depth that would normally constitute a completed work and, in general, the latter half of this study loses focus as it tries to cover "too much ground" and is disconnected from the earlier experiments. After reading all of this, I came away with the impression that this really is two stories merged together, and one alternative strategy would be to really consolidate the mechanisms up to Figure 3 and 4, then split Figures 5 and 6 into a distinct study. This is a tough call, as it is also attractive to present the complete series of events from compaction to relaxation back to compaction, even if many details and nuances remain unexplored. If the authors want to keep all of the data in a single study, then I would suggest increasing degree of clarity of the intermediate steps of the mechanism as that remains, to me, the weakest part of the work. They also need to carry through the main DDB2 narrative through the final experiments.

Specific Comments:

1. Figure 1. The impact of SUV39H1/2 depletion on survival after UV appears to only be significant

(and is still modest) after an exceptionally high UVC dose (12 J/m²) with no significant difference at any lower dose. This isn't acknowledged in the description of the results, but should be. Saying that "Loss of heterochromatin integrity thus correlates with reduced cell viability following UV damage" based on that outcome is a bit weak.

2. One of the biggest concerns I have with this work is that adding Hoechst dye (an intercalating dye) before localized UV microirradiation is a classic way to induce DNA double strand breaks for live cell imaging experiments, although this approach has somewhat fallen out of favour due to the abnormal DNA damage response(s) that are elicited due to the combination of intercalation, high dose and other factors only partly understood. The numerous nuances and issues with the approach have been articulated in: <https://academic.oup.com/nar/article/37/9/e68/1163433>, <https://www.nature.com/articles/s41598-018-26307-6>, and others, with the somewhat famous artefacts observed for TRF2 using this technique in the mid-2000s

(<https://pubmed.ncbi.nlm.nih.gov/15665826/>, <https://pubmed.ncbi.nlm.nih.gov/17534357/>)

I recognize that the approach in this study uses UVC and not UVA (which is the classic way of combining Hoechst and UV to get DSBs), but using UVC also has some issues as the authors of this work will be aware of (as they've reviewed this type of work before, see link below) with particular emphasis on issues in monitoring green signal after such an approach (as done in Figure 1C).

<https://www.ncbi.nlm.nih.gov/pmc/articles/PMC6392149/>

Looking at the 266 nm UVC laser conditions, it is not clear to me whether these (for this specific cell model) have issues of false positive / altered green, cyan or red signal and/or will be generating any DSBs and other "abnormal" damage, or if it will be actually generating something similar to high dose UVC lesions that mirror the global UVC methodology using the low pressure mercury lamps used in other experiments. There are a few ways to address this (although non perfect, I admit), including relative CPD levels, relative gammaH2AX, 53BP1 and related DSB-specific marker staining, examining for ATM activation, etc. I think it is important to establish exactly what spectrum and type of damage is taking place using this methodology for the readers to make sense of the outcomes. Otherwise, it is possible that a proportion of these phenomena may be attributable to artefacts of the method or a DSB response more similar to ionizing radiation exposure rather than UVC damage. To be clear, I'm not saying this is the case, I'm just asking that it be addressed with methodology controls so we are more certain.

3. Related to the above (method concerns) is the issue of relative UV damage dose. It isn't clear how the localized UVC irradiation method using the UVC laser compares to the global exposure approaches in terms of J/m². This is relevant to ascertain how comparable outcomes of the two approaches are throughout the study. The authors should comment and show some evidence for the number of UV lesions (and what types) and their relative density between each technique for this specific cell system(s).

4. Figure 2. DDB2 forms a heterodimer with DDB1 (<https://pubmed.ncbi.nlm.nih.gov/16223728/>) and complexes with CUL4A to form a UB-ligase, but normally occupies UV induced lesions for a very short period of time (<https://pubmed.ncbi.nlm.nih.gov/17635991/>). Its chromatin remodelling action is also understood to occur via ALC1

(<https://pubmed.ncbi.nlm.nih.gov/23045548/>). In the context of the experiments described in this manuscript, have the authors confirmed that there is sufficient DDB1 expressed in NIH3T3 cells to form the heterodimer at a stoichiometric level? What are the implications of a forced-tethering of DDB2 to chromatin for prolonged periods, where normally it has a limited residence time at UV damage? Is the relaxation effect observed on chromocentres dependent on ALC1/CHD1L? The experiments in Figure 2 would be strengthened by addressing the effects in the context of established DDB2 molecular mechanism for chromatin remodelling at UV lesions. At minimum, the dependence on ALC1 and interactions with DDB1 and/or CUL4A should be explored. Given that the chromatin remodelling action of DDB2 has been at least partially mapped out, the statement (p 9) that "Given that DDB2 does not harbour known chromatin remodelling activity or motifs, we hypothesized that it controlled chromatin compaction indirectly" seems odd, in that it ignores that we already "know" that DDB2 chromatin relaxation is indirect (and how) in the context of UV induced DNA damage chromatin dynamics.

5. Figure 3. PARP activity directed against linker Histone H1 has been suggested as a key mechanism of displacement (<https://pubmed.ncbi.nlm.nih.gov/26559976/>). In the context of the experiments shown in Figure 3, and using UVC damage (notwithstanding concerns of the method articulated above), does PARP activity also play a role in the action of DDB2, given that DDB2 is known to be PARylated itself? This line of investigation is, similar to the ALC1 question from

previous point, necessary to consolidate the intermediate steps of the overall mechanism shown in this work. PARP1/2 depletion or PARP inhibitors could be used to address this.

6. Figure 4, in contrast to the title of this section in the results, strictly speaking, the authors do not measure HC vs. EC UV damage repair over time in these experiments. That would require monitoring CPD signal over time in each region and quantifying the outcomes. Most of these experiments indicate that UV damage repair factors can access the site, but not necessarily do they confirm that are they actually functioning. For example, we know that most NHEJ factors do accumulate in heterochromatic DSBs, but that they cannot actually repair damage until chromatin is remodelled. It's an important thing to distinguish. I would like to see some indication of relative CPD resolution HC over time in this experimental context, and whether DDB2 status (i.e. chromatin relaxation in HC) influences this to a significant extent. As in, does relaxation of the chromatin "matter" to the speed or outcome of UV lesion repair in heterochromatin over time?

7. As indicated above, the data in Figures 5 and 6 could encompass a study in and of themselves. Related to the concern in #6 above, however, it is not clear how the fresh deposition of *heterochromatic* histones and H3K9me3 maintenance relates to the outcome and/or kinetics of UV lesion repair in heterochromatin. So, in line with the comment in #6, what happens to CPD resolution over time if the action of DAXX or HIRA are perturbed, or what happens to DAXX or HIRA action if CPD resolution is blocked? The authors find that ATRX is needed for DAXX recruitment, but do not appear to address whether DDB2 or its chromatin remodelling activity are involved in this, which seems odd in that this is the primary focus of the study up to this point. Altogether, the connections between the earlier experiments shown in this study and these latter experiments starts to 'break down' by this point (Fig 5). By Figure 6, DDB2 and previous lines of investigation are more or less dropped altogether... this is a weakness and could be strengthened if the influence of DDB2-dependent (UV-induced heterochromatin relaxation dynamics) pathways on the whole mechanism are considered and addressed all the way through the work.

Reviewer #3 (Remarks to the Author):

Fortuny et al., Imaging the response to DNA damage in heterochromatin domains reveals core principles of heterochromatin maintenance

In this manuscript, Fortuny et al., analyze the UV DNA damage response in heterochromatin. The authors identify a striking decompaction and recompaction of the heterochromatin domain upon UV irradiation of mouse cells, which the authors find to be regulated by the UV DNA damage sensor DDB2. In addition, the authors follow histone H3 deposition upon UV damage and identify the histone chaperone HIRA as the main driver of histone H3.3 deposition. Finally, the authors find a specific role for the histone methyltransferase SETDB1 in promoting the maintenance of the heterochromatin histone mark H3K9me3 following UV DNA damage exposure of heterochromatic regions.

This manuscript creates an in-depth overview of the UV DNA damage response in heterochromatic regions, which will be interesting for a broad audience. To my knowledge, this is the first time someone has performed such a detailed characterization of the heterochromatic UV DNA damage response, and the experiments in this manuscript will likely lay the foundation for more exciting, future work on this topic. More importantly, the experiments in this manuscript allow the authors to draw conclusions on the specific timing of events, which I think is essential to properly identify the responses associated with such a dynamic process as the (hetero)chromatin response to DNA damage.

In general the experiments are of high quality and the conclusions drawn by the authors are supported by their experiments. I therefore recommend publication of this manuscript in Nature Communications. Below I outline some minor concerns that I think should be addressed before publication.

Minor comments

- Page 2; rewrite this sentence in the abstract: (see bold) "...We unveil a central role for the methyltransferase SETDB1 in the maintenance of heterochromatic histone marks after UV, SETDB1 coordinating histone methylation with new histone deposition in damaged heterochromatin, thus

protecting cells from genome instability...'

- Fig.S2b. The FISH data in Fig.S2b needs a quantification similar to the quantification in Fig.S2a.
- Fig.1/6. The authors claim that H3K9me3 is 'maintained' following UV DNA damage. The fact that the authors see a decrease in H3K9me3 in the absence of SETDB1 (Fig.6e), indicates that it is not a static 'maintenance' mechanism, but rather a dynamic process of removing histone marks/repositioning old/new histones/methylating new/old histones. To address the exact timing of H3K9 (de)methylation events, I would suggest to determine H3K9me3 levels at different timepoints following UV DNA damage induction. The authors have the perfect system to test this and assess changes in H3K9me3 levels during for example decompaction/recompaction instead of focusing on only one time-point (e.g. 60min in Fig.1C or 90min in Fig.6d, e).
- Fig. 5d. It is unclear to me why the authors only analyze incorporation of newly synthesized H3.3 following UV DNA damage induction in heterochromatin, and do not analyze H3.1 incorporation. To create a more complete overview of the UV DNA damage response and its associated H3.1/H3.3 deposition, I think the manuscript would benefit from including H3.1 SNAP experiments as well, and assessing the impact of CAF-1 knockdown on this deposition.
- Previously it was described that heterochromatin has a delayed NER response when compared to euchromatin (Han C et al., Carcinogenesis 2016/Adar PNAS 2016/Hu PNAS 2016). However, based on the experiments shown, the authors here do not identify major differences between eu- and heterochromatin in terms of recruitment of NER repair proteins as well as H3.3 deposition. To address differences in UV repair kinetics between eu- and heterochromatin, it would be informative if the authors could analyze the specific timing of events; e.g. are there differences in timing of DDB2/NER repair protein recruitment or histone H3 deposition between eu- and heterochromatin?
- Fig. 6f. The authors find that SETDB1 is responsible for the increase in H3K9me3 following UV DNA damage induction. To identify the relevance of SETDB1 dependent H3K9me3 methylation in heterochromatin repair, the authors perform a rather indirect measure of this functional impact by knocking down SETDB1 and assessing micronuclei accumulation (-/+ UV) (Fig.6f). However, I think it would improve this manuscript if the authors could perform more direct experiments on the role of SETDB1 in recovery from UV DNA damage repair. As such, I would suggest to knockdown SETDB1 and analyze whether this has any impact on heterochromatin recompaction and/or removal of UV NER repair proteins.
- Fig.6. The authors mainly focus on the hypothesis that SETDB1 is the enzyme responsible for the increase in H3K9me3 at UV DNA damage sites, and that SUV39H1/H2 solely promotes H3K9me3 of parental histones, not newly deposited histones. However, with their current experiments, the authors cannot rule out a role for SUV39H1/H2 in methylating H3K9me3 at the UV damage sites. Although I understand that the authors cannot directly test the role of SUV39H1/H2 in methylating newly deposited histones in their assay (since its loss will remove almost all H3K9me3 in the nucleus), I do think the authors should include some experiments to test the role of SUV39H1/H2 in methylating H3K9me3 at the DNA damage site. For example, the authors only determine the impact of SUV39H1/H2 loss or CAF-1/H3.3 loss on SETDB1 recruitment to UV DNA damage sites (Fig.6c). However, the reciprocal experiments are missing in which the authors determine the effect of loss of SETDB1 or CAF-1/H3.3 on SUV39H1/H2 recruitment (Fig.S6A) to UV damage sites. These experiments would give more insight into the recruitment and roles of these different H3K9me3 methyltransferases at the DNA damage sites and thereby strengthen the authors' conclusions.

Point-by-point responses to reviewers' comments:

We thank all three reviewers for their thorough evaluation of our manuscript and for constructive criticism. Their insightful comments helped to improve our manuscript and strengthen our conclusions. The main additions to our work are as follows:

- We provide further **mechanistic insights** into heterochromatin decompaction following UV damage showing no dependency on UV damage processing, cell cycle stage, PARP-dependent chromatin remodeling. We also further investigate the mechanistic links between DDB2 and other players in heterochromatin maintenance and the interplay between H3K9 trimethylating enzymes.
- We better characterize the **UVC laser damage methodology** that we employ in our study.
- We perform **kinetic analyses** for DDB2 recruitment to damaged heterochromatin vs. euchromatin, and for H3K9me3 level changes post UV.

Reviewer #1:

The authors have investigated the events occurring at heterochromatin following UV damage. In particular they found that damaged heterochromatin undergoes decompaction and then recompaction. Histone modifications and transcriptional silencing are maintained. While DPB2 influences heterochromatin compaction, SETDB1 is required to maintain histone marks following UV damage.

General comments

They used immunofluorescent techniques in combination with UVC laser coupled to confocal microscope to follow the events at the heterochromatin in engineered NIH/3T3 mouse embryonic fibroblasts. The experimental system is brave and original. Overall the paper is interesting and the findings are potentially relevant. However, there are crucial points that must be addressed.

Specific comments

1. The first chapter of the results section should describe the experimental system. This is important as the entire paper is based on this approach. In particular the levels of GFP-hDDB2 must be shown.

We have subdivided the first result section into two parts so that the first one describes the experimental system (p. 5 of our revised manuscript). We now show the levels of GFP-hDDB2 in NIH/3T3 cells compared to DDB2 expression in human MCF7 cells (Supplementary Fig. 1b).

2. Is the decompaction/compaction influenced by NER processing/repair? In particular is the decompaction dependent on NER-mediated DNA processing? This is an important control to rule out other effects caused by laser treatment (such as protein/membrane damage).

To address whether heterochromatin decompaction was influenced by UV damage processing/repair, downstream of DDB2, we knocked down the NER factor XPC. This affected the recruitment of repair factors involved in UV damage processing, including XPB, but did not prevent the decompaction of damaged heterochromatin (Supplementary Fig. 4e), showing that heterochromatin decompaction depends on

DDB2 but not on downstream steps of UV damage processing and repair. Furthermore, the decompaction of heterochromatin observed upon dCas9-mediated tethering of DDB2 occurred without the recruitment of UV damage processing factor XPB (Supplementary Fig. 4f).

The dependency on DDB2 does not point to protein or membrane damage triggering heterochromatin decompaction at sites of laser damage. To further evaluate other possible effects of the UVC laser treatment, we performed additional controls, presented in a new Supplementary Fig 2, ruling out any bleaching of the Hoechst signal and any switch from blue to green by the UVC laser and also showing that the combination of UVC laser and Hoechst does not trigger a DSB response.

3. Does the cell cycle influence heterochromatin decompaction? In particular it would be important to see G1 cells.

We have observed the decompaction of damaged heterochromatin in all the cells that we have analyzed. A cell cycle dependency of the process is therefore unlikely. Our FACS analysis shows that most NIH/3T3 cells in the population are in G1 (Supplementary Fig. 1d), supporting the idea that G1 cells display heterochromatin decompaction upon UV damage. To strengthen this point, we now show that decompaction of damaged heterochromatin occurs in and outside S-phase (Supplementary Fig. 3c).

Reviewer #2:

Fortuny et al NCB Review

In their manuscript, “Imaging the response to DNA damage in heterochromatin domains reveals core principles of heterochromatin maintenance”, Fortuny et al. explore the molecular mechanism of ultraviolet C light induced DNA damage repair within regions of constitutive heterochromatin using a targeted damage approach in an NIH3T3 murine and MCF7 human cancer-derived cell model. I found this study very well written, and am broadly supportive of the work as experiments are high quality, the topic is of broad interest, and they are asking mechanistic question. I do, however, have numerous concerns relating to the methodology that require clarification and/or added controls (see below).

General comment on all the data that lead to the final model shown in Figure 7. The authors present data for each step in a process for heterochromatin relaxation and recompacting during UV DNA damage responses. This is good, but I note that the details of each step in the mechanism (especially intermediate steps, i.e. the data in Figure 3-4) aren't really studied with the degree of depth that would normally constitute a completed work and, in general, the latter half of this study loses focus as it tries to cover “too much ground” and is disconnected from the earlier experiments. After reading all of this, I came away with the impression that this really is two stories merged together, and one alternative strategy would be to really consolidate the mechanisms up to Figure 3 and 4, then split Figures 5 and 6 into a distinct study. This is a tough call, as it is also attractive to present the complete series of events from compaction to relaxation back to compaction, even if many details and nuances remain unexplored. If the authors want to keep all of the data in a single study, then I would suggest increasing degree of clarity of the intermediate steps of the mechanism as that remains, to me, the weakest part of the work.

They also need to carry through the main DDB2 narrative through the final experiments.

To address this point, we now show that SETDB1 is recruited to damaged heterochromatin in a DDB2-dependent manner (Supplementary Fig. 7a).

Specific Comments:

1. Figure 1. The impact of SUV39H1/2 depletion on survival after UV appears to only be significant (and is still modest) after an exceptionally high UVC dose (12 J/m²) with no significant difference at any lower dose. This isn't acknowledged in the description of the results, but should be. Saying that “Loss of heterochromatin integrity thus correlates with reduced cell viability following UV damage” based on that outcome is a bit weak.

We have rephrased our conclusion accordingly (p. 6)

2. One of the biggest concerns I have with this work is that adding Hoechst dye (an intercalating dye) before localized UV microirradiation is a classic way to induce DNA double strand breaks for live cell imaging experiments, although this approach has somewhat fallen out of favour due to the abnormal DNA damage response(s) that are elicited due to the combination of intercalation, high dose and other factors only partly understood. The numerous nuances and issues with the approach have been articulated in: <https://academic.oup.com/nar/article/37/9/e68/1163433>, <https://www.nature.com/articles/s41598-018-26307-6>, and others, with the somewhat famous artefacts observed for TRF2 using this technique in the mid-2000s

(<https://pubmed.ncbi.nlm.nih.gov/15665826/>,
<https://pubmed.ncbi.nlm.nih.gov/17534357/>)

I recognize that the approach in this study uses UVC and not UVA (which is the classic way of combining Hoechst and UV to get DSBs), but using UVC also has some issues as the authors of this work will be aware of (as they've reviewed this type of work before, see link below) with particular emphasis on issues in monitoring green signal after such an approach (as done in Figure 1C).
<https://www.ncbi.nlm.nih.gov/pmc/articles/PMC6392149/>

Looking at the 266 nm UVC laser conditions, it is not clear to me whether these (for this specific cell model) have issues of false positive / altered green, cyan or red signal and/or will be generating any DSBs and other "abnormal" damage, or if it will be actually generating something similar to high dose UVC lesions that mirror the global UVC methodology using the low pressure mercury lamps used in other experiments. There are a few ways to address this (although non perfect, I admit), including relative CPD levels, relative gammaH2AX, 53BP1 and related DSB-specific marker staining, examining for ATM activation, etc. I think it is important to establish exactly what spectrum and type of damage is taking place using this methodology for the readers to make sense of the outcomes. Otherwise, it is possible that a proportion of these phenomena may be attributable to artefacts of the method or a DSB response more similar to ionizing radiation exposure rather than UVC damage. To be clear, I'm not saying this is the case, I'm just asking that it be addressed with methodology controls so we are more certain.

While the combination of Hoechst and UVA laser is commonly used to trigger DNA double-strand breaks (DSBs) among other DNA lesions, Hoechst plus UVC does not elicit a DSB response, as shown by the lack of recruitment of the DSB repair factor KU70 (Supplementary Fig. 2c). In line with this, the dependency on DDB2 that we uncover does not point to a DSB response, DDB2 being involved in NER and BER pathways only. To further evaluate other possible effects of the UVC laser treatment, we performed additional controls, ruling out any bleaching of the Hoechst signal and any switch from blue to green by the UVC laser (Supplementary Fig. 2a-b).

3. Related to the above (method concerns) is the issue of relative UV damage dose. It isn't clear how the localized UVC irradiation method using the UVC laser compares to the global exposure approaches in terms of J/m². This is relevant to ascertain how comparable outcomes of the two approaches are throughout the study. The authors should comment and show some evidence for the number of UV lesions (and what types) and their relative density between each technique for this specific cell system(s).

We have compared UVC laser-induced damage to local damage induced by the UVC lamp based on CPD staining in our murine cellular model (Supplementary Fig. 2d). The UVC dose delivered at the site of laser micro-irradiation can thus be estimated at 600 J/m², which generates on average 1 pyrimidine dimer every nucleosome (1 UV lesion/ 150 bp) in 2% of the nuclear volume, while global UV irradiation with the lamp at 10 J/m² generates 60 times less damage (1 UV lesion/ 10 kb) in the whole nuclear volume. The expected damage load for a given UV dose is based on van Zeeland et al, Mutat Res, 1981. We have included this information in the method section of our revised manuscript (p. 24).

4. Figure 2. DDB2 forms a heterodimer with DDB1 (<https://pubmed.ncbi.nlm.nih.gov/16223728/>) and complexes with CUL4A to form a

UB-ligase, but normally occupies UV induced lesions for a very short period of time (<https://pubmed.ncbi.nlm.nih.gov/17635991/>). Its chromatin remodelling action is also understood to occur via ALC1 (<https://pubmed.ncbi.nlm.nih.gov/23045548/>). In the context of the experiments described in this manuscript, have the authors confirmed that there is sufficient DDB1 expressed in NIH3T3 cells to form the heterodimer at a stoichiometric level? What are the implications of a forced-tethering of DDB2 to chromatin for prolonged periods, where normally it has a limited residence time at UV damage? Is the relaxation effect observed on chromocentres dependent on ALC1/CHD1L? The experiments in Figure 2 would be strengthened by addressing the effects in the context of established DDB2 molecular mechanism for chromatin remodelling at UV lesions. At minimum, the dependence on ALC1 and interactions with DDB1 and/or CUL4A should be explored. Given that the chromatin remodelling action of DDB2 has been at least partially mapped out, the statement (p 9) that “Given that DDB2 does not harbour known chromatin remodelling activity or motifs, we hypothesized that it controlled chromatin compaction indirectly” seems odd, in that it ignores that we already “know” that DDB2 chromatin relaxation is indirect (and how) in the context of UV induced DNA damage chromatin dynamics.

We now show by western blot that DDB1 levels are slightly increased in murine cells ectopically expressing GFP-hDDB2 (Supplementary Fig. 1b). Additionally, we show that ectopically expressed GFP-hDDB2 pulls down endogenous DDB1 in NIH/3T3 cells (Supplementary Fig. 1c).

The forced tethering of DDB2 to chromatin indeed does not mirror the more transient binding of DDB2 to chromatin observed during UV damage repair but it is the only way to test the impact of DDB2 on chromatin decompaction independently of DNA damage.

We have knocked down the chromatin remodeler ALC1 to assess its contribution to the decompaction of UV-damaged heterochromatin. We still observed heterochromatin decompaction in these conditions, consistent with our previous findings in human chromatin (Adam et al., Mol Cell 2016). However, due to the lack of specific antibodies to detect ALC1 by immunofluorescence in mouse cells (we tested three different antibodies), we could not reliably control that heterochromatin decompaction was observed in cells with reduced ALC1 levels. We thus decided not to include these data in our revised manuscript. Nevertheless, as explained in point 5 below, we provide evidence that PARP activity is dispensable for the decompaction of UV-damaged heterochromatin. It is therefore unlikely that the PARP-dependent remodeler ALC1 plays a role in this process.

5. Figure 3. PARP activity directed against linker Histone H1 has been suggested as a key mechanism of displacement (<https://pubmed.ncbi.nlm.nih.gov/26559976/>). In the context of the experiments shown in Figure 3, and using UVC damage (notwithstanding concerns of the method articulated above), does PARP activity also play a role in the action of DDB2, given that DDB2 is known to be PARylated itself? This line of investigation is, similar to the ALC1 question from previous point, necessary to consolidate the intermediate steps of the overall mechanism shown in this work. PARP1/2 depletion or PARP inhibitors could be used to address this.

This is an important point considering the established links between PARP and H1 eviction on one hand (Strickdafen et al., J Biol Chem 2016) and PARP and DDB2 on the other hand (Pines et al., J Cell Biol 2012; Robu et al., PNAS 2013). Upon PARP inhibition, we still observe the decompaction of UV-damaged heterochromatin, ruling out a role of PARylation in this process (Supplementary Fig. 4a).

6. Figure 4, in contrast to the title of this section in the results, strictly speaking, the authors do not measure HC vs. EC UV damage repair over time in these experiments. That would require monitoring CPD signal over time in each region and quantifying the outcomes. Most of these experiments indicate that UV damage repair factors can access the site, but not necessarily do they confirm that are they actually functioning. For example, we know that most NHEJ factors do accumulate in heterochromatic DSBs, but that they cannot actually repair damage until chromatin is remodelled. It's an important thing to distinguish. I would like to see some indication of relative CPD resolution over time in this experimental context, and whether DDB2 status (i.e. chromatin relaxation in HC) influences this to a significant extent. As in, does relaxation of the chromatin "matter" to the speed or outcome of UV lesion repair in heterochromatin over time?

Slower CPD resolution in heterochromatin and the importance of DDB2 for CPD repair in heterochromatin have been shown previously by the Sancar and Wani groups (Adar et al, NAS 2016; Han et al, Carcinogenesis 2016), as discussed on p.17 of our revised manuscript, suggesting that DDB2-dependent relaxation of heterochromatin indeed promotes UV damage repair.

7. As indicated above, the data in Figures 5 and 6 could encompass a study in and of themselves. Related to the concern in #6 above, however, it is not clear how the fresh deposition of *heterochromatic* histones and H3K9me3 maintenance relates to the outcome and/or kinetics of UV lesion repair in heterochromatin. So, in line with the comment in #6, what happens to CPD resolution over time if the action of DAXX or HIRA are perturbed, or what happens to DAXX or HIRA action if CPD resolution is blocked?

We now examine the impact of DAXX and HIRA on GFP-DDB2 removal from damaged heterochromatin, ruling out a contribution of these chaperones to UV damage repair in these chromatin domains (Supplementary Fig. 6f).

The authors find that ATRX is needed for DAXX recruitment, but do not appear to address whether DDB2 or it's chromatin remodelling activity are involved in this, which seems odd in that this is the primary focus of the study up to this point.

As shown in Supplementary Fig. 6e, DDB2 is required for DAXX recruitment to damaged heterochromatin.

Altogether, the connections between the earlier experiments shown in this study and these latter experiments starts to 'break down' by this point (Fig 5). By Figure 6, DDB2 and previous lines of investigation are more or less dropped altogether... this is a weakness and could be strengthened if the influence of DDB2-dependent (UV-induced heterochromatin relaxation dynamics) pathways on the whole mechanism are considered and addressed all the way through the work.

To better connect early and late experiments in our study, we now show that SETDB1 is recruited to damaged heterochromatin in a DDB2-dependent manner (Supplementary Fig. 7a). Reciprocally, we assess the importance of SETDB1 in UV damage repair by monitoring GFP-DDB2 release from damaged heterochromatin, ruling out a contribution of SETDB1 to UV damage repair in these chromatin domains (Supplementary Fig. 7d).

Reviewer #3:

Fortuny et al., Imaging the response to DNA damage in heterochromatin domains reveals core principles of heterochromatin maintenance

In this manuscript, Fortuny et al., analyze the UV DNA damage response in heterochromatin. The authors identify a striking decompaction and recompaction of the heterochromatin domain upon UV irradiation of mouse cells, which the authors find to be regulated by the UV DNA damage sensor DDB2. In addition, the authors follow histone H3 deposition upon UV damage and identify the histone chaperone HIRA as the main driver of histone H3.3 deposition. Finally, the authors find a specific role for the histone methyltransferase SETDB1 in promoting the maintenance of the heterochromatin histone mark H3K9me3 following UV DNA damage exposure of heterochromatic regions.

This manuscript creates an in-depth overview of the UV DNA damage response in heterochromatic regions, which will be interesting for a broad audience. To my knowledge, this is the first time someone has performed such a detailed characterization of the heterochromatic UV DNA damage response, and the experiments in this manuscript will likely lay the foundation for more exciting, future work on this topic. More importantly, the experiments in this manuscript allow the authors to draw conclusions on the specific timing of events, which I think is essential to properly identify the responses associated with such a dynamic process as the (hetero)chromatin response to DNA damage.

In general the experiments are of high quality and the conclusions drawn by the authors are supported by their experiments. I therefore recommend publication of this manuscript in Nature Communications. Below I outline some minor concerns that I think should be addressed before publication.

Minor comments

- Page 2; rewrite this sentence in the abstract: (see bold) "...We unveil a central role for the methyltransferase SETDB1 in the maintenance of heterochromatic histone marks after UV, SETDB1 coordinating histone methylation with new histone deposition in damaged heterochromatin, thus protecting cells from genome instability..."

We have rephrased this sentence accordingly (p. 2).

- Fig.S2b. The FISH data in Fig.S2b needs a quantification similar to the quantification in Fig.S2a.

We now provide a 3D quantification for the FISH data (Supplementary Fig. 3b).

- Fig.1/6. The authors claim that H3K9me3 is 'maintained' following UV DNA damage. The fact that the authors see a decrease in H3K9me3 in the absence of SETDB1 (Fig.6e), indicates that it is not a static 'maintenance' mechanism, but rather a dynamic process of removing histone marks/repositioning old/new histones/methylating new/old histones. To address the exact timing of H3K9 (de)methylation events, I would suggest to determine H3K9me3 levels at different timepoints following UV DNA damage induction. The authors have the perfect system to test this and assess changes in H3K9me3 levels during for example decompaction/recompaction instead of focusing on only one time-point (e.g. 60min in

Fig.1C or 90min in Fig.6d, e).

As suggested by the Reviewer, we have analyzed by western blot H3K9me3 levels at different time points following UV damage including time points before and after 60-90 min. This revealed a gradual increase in H3K9me3 levels up to 3h post UV without any detectable drop at early time points (Supplementary Fig. 3g), thus supporting the idea that this heterochromatin mark is maintained, and even slightly increased post UV, rather than removed and subsequently re-established.

- Fig. 5d. It is unclear to me why the authors only analyze incorporation of newly synthesized H3.3 following UV DNA damage induction in heterochromatin, and do not analyze H3.1 incorporation. To create a more complete overview of the UV DNA damage response and its associated H3.1/H3.3 deposition, I think the manuscript would benefit from including H3.1 SNAP experiments as well, and assessing the impact of CAF-1 knockdown on this deposition.

We originally intended to monitor new H3.1 deposition as done for new H3.3 but technical issues prevented this analysis. New H3.1 levels were indeed much lower than new H3.3. They were difficult to visualize at replication foci in our murine cell line model and therefore impossible to detect at damage sites. This is why we can only provide the recruitment of the H3.1 chaperone CAF-1 to damaged heterochromatin. Nevertheless, the focus on new H3.3 deposition in our study is very relevant considering that this histone variant was shown to be deposited in pericentric heterochromatin domains in the absence of DNA damage (Drané et al., Genes Dev 2010).

- Previously it was described that heterochromatin has a delayed NER response when compared to euchromatin (Han C et al., Carcinogenesis 2016/Adar PNAS 2016/Hu PNAS 2016). However, based on the experiments shown, the authors here do not identify major differences between eu- and heterochromatin in terms of recruitment of NER repair proteins as well as H3.3 deposition. To address differences in UV repair kinetics between eu- and heterochromatin, it would be informative if the authors could analyze the specific timing of events; e.g. are there differences in timing of DDB2/NER repair protein recruitment or histone H3 deposition between eu- and heterochromatin?

We have compared the kinetics of DDB2 recruitment to damaged euchromatin and heterochromatin (Fig. 4a). We observe that DDB2 recruitment to UV damage is not delayed in heterochromatin.

- Fig. 6f. The authors find that SETDB1 is responsible for the increase in H3K9me3 following UV DNA damage induction. To identify the relevance of SETDB1 dependent H3K9me3 methylation in heterochromatin repair, the authors perform a rather indirect measure of this functional impact by knocking down SETDB1 and assessing micronuclei accumulation (-/+ UV) (Fig.6f). However, I think it would improve this manuscript if the authors could perform more direct experiments on the role of SETDB1 in recovery from UV DNA damage repair. As such, I would suggest to knockdown SETDB1 and analyze whether this has any impact on heterochromatin recompaction and/or removal of UV NER repair proteins.

To assess the role of SETDB1 in UV damage repair progression, we have monitored GFP-DDB2 release from damaged heterochromatin, ruling out a direct contribution of SETDB1 to UV damage repair in these chromatin domains (Supplementary Fig. 7d). The impact of SETDB1 on micronuclei formation post UV thus reflects a role of

SETDB1 in maintaining pericentric heterochromatin integrity rather than a contribution of SETDB1 to DNA repair *per se*.

- Fig.6. The authors mainly focus on the hypothesis that SETDB1 is the enzyme responsible for the increase in H3K9me3 at UV DNA damage sites, and that SUV39H1/H2 solely promotes H3K9me3 of parental histones, not newly deposited histones. However, with their current experiments, the authors cannot rule out a role for SUV39H1/H2 in methylating H3K9me3 at the UV damage sites.

Although I understand that the authors cannot directly test the role of SUV39H1/H2 in methylating newly deposited histones in their assay (since its loss will remove almost all H3K9me3 in the nucleus), I do think the authors should include some experiments to test the role of SUV39H1/H2 in methylating H3K9me3 at the DNA damage site. For example, the authors only determine the impact of SUV39H1/H2 loss or CAF-1/H3.3 loss on SETDB1 recruitment to UV DNA damage sites (Fig.6c). However, the reciprocal experiments are missing in which the authors determine the effect of loss of SETDB1 or CAF-1/H3.3 on SUV39H1/H2 recruitment (Fig.S6A) to UV damage sites. These experiments would give more insight into the recruitment and roles of these different H3K9me3 methyltransferases at the DNA damage sites and thereby strengthen the authors' conclusions.

We now show that SETDB1 loss does not impact SUV39H1 recruitment to UV damaged heterochromatin (Supplementary Fig. 7b)

REVIEWERS' COMMENTS

Reviewer #1 (Remarks to the Author):

I am pleased with the revised version

Reviewer #2 (Remarks to the Author):

In their revised manuscript, "Imaging the response to DNA damage in heterochromatin domains reveals core principles of heterochromatin maintenance", Fortuny et al. present a series of refinements and controls that substantially improve what was already, in my view, a strong study. I commend the authors on the additional controls (especially Figure S2) and text edits, as well as the new lines of investigation (such as determining role of PARP, etc.) that add to the major claims of the paper and really enhance confidence in the fundamentals of the core methods.

I now consider this a sufficiently consolidated story and, given the massive amount of supplementary and primary data that collectively address any major issues, am not inclined to ask for any further experiments or revisions. This work is ready for the larger scientific community at large to view it.

Reviewer #3 (Remarks to the Author):

Fortuny et al., have revised their manuscript 'Imaging the response to DNA damage in heterochromatin domains reveals core principles of heterochromatin maintenance' according to the reviewers' comments. I am satisfied with their responses and newly incorporated data and have no further comments.

Response to reviewers' comments

Reviewer #1:

I am pleased with the revised version

Reviewer #2:

In their revised manuscript, "Imaging the response to DNA damage in heterochromatin domains reveals core principles of heterochromatin maintenance", Fortuny et al. present a series of refinements and controls that substantially improve what was already, in my view, a strong study. I commend the authors on the additional controls (especially Figure S2) and text edits, as well as the new lines of investigation (such as determining role of PARP, etc.) that add to the major claims of the paper and really enhance confidence in the fundamentals of the core methods.

I now consider this a sufficiently consolidated story and, given the massive amount of supplementary and primary data that collectively address any major issues, am not inclined to ask for any further experiments or revisions. This work is ready for the larger scientific community at large to view it.

Reviewer #3:

Fortuny et al., have revised their manuscript 'Imaging the response to DNA damage in heterochromatin domains reveals core principles of heterochromatin maintenance' according to the reviewers' comments. I am satisfied with their responses and newly incorporated data and have no further comments.

We thank all three reviewers for their positive evaluation of our revised manuscript. No further changes were requested.